# Method for predicting dynamic current-carrying capacity of transmission lines by integrating improved VMD and time-varying ensemble model

Siyu Yang ⓘ *, Wubang Hao

School of Electrical and Information Engineering, Yunnan Minzu University, Kunming, China

* 1762890729@qq.com

## Abstract

Accurately predicting the dynamic thermal rating of power lines is the core of overcoming the bottleneck in the engineering application of line dynamic capacity – increase technology. In response to the limitations of DTR data decomposition caused by the existing methods relying on manual experience to set the hyperparameters of the Variational Mode Decomposition algorithm, and the problems of single models, such as their inability to accurately extract multi – scale time – series features of DTR data and poor generalization ability, this paper proposes a transmission line DTR prediction method driven by an improved VMD and based on a time – varying multi – model ensemble. First, by constructing a fitness function based on the minimum envelope entropy of DTR data components, a mathematical mapping relationship between the hyperparameter space and the VMD decomposition effect is established. This transforms the traditional experience – oriented hyperparameter selection into a quantifiable optimization problem for solution. Subsequently, the global search ability of the slime mold algorithm is utilized to iteratively search for the optimal hyperparameters of VMD in the hyperparameter space. Second, by integrating the dynamic recursive characteristics of Elman and the multi – scale convolutional feature extraction advantages of TCN, a time – varying ensemble model is constructed to accurately extract the long – and short – term time – series features of DTR data. In particular, a dynamic weighting mechanism based on grey relational coefficients is designed. By quantifying the local correlation between the predicted values of the Elman model and the TCN model and the real values point – by – point, the weights can be adaptively allocated, effectively solving the problem of insufficient generalization ability of single models. Experimental results show that the proposed method maintains a prediction accuracy of over 95% in different datasets, which is 5.7% higher than that of the best traditional methods. It provides more theoretical support for the safe capacity increase of power lines and has significant engineering application value.

**Data availability statement:** All relevant data are within the paper and its Supporting Information files.

**Funding:** The author(s) received no specific funding for this work.

**Competing interests:** The authors have declared that no competing interests exist.

## 1. Introduction

The safe current-carrying capacity of transmission lines is a key factor restricting the transmission efficiency of power grids, and the accuracy of its assessment directly affects the operational reliability of lines and resource utilization efficiency [1]. As the primary method for evaluating the safe current-carrying capacity of transmission lines, the traditional Static Thermal Rating (STR) method generally establishes current-carrying models under the constraint of conductor temperature rise based on extreme meteorological conditions [2]. While this method can ensure the operational reliability of transmission lines under extreme meteorological conditions, it has significant limitations in practical operation. When various meteorological parameters in the actual operating environment are better than those of extreme conditions, STR leads to a conservative estimation of transmission line capacity, resulting in inefficient utilization of transmission corridor resources [3,4].

To overcome the conservative estimation of STR, the dynamic capacity-increase technology for transmission lines has emerged. The core goal of this technology is to establish a Dynamic Thermal Rating (DTR) prediction model by real-time coupling of various meteorological conditions and conductor thermal characteristics, thereby maximizing transmission capacity on the premise of ensuring the safe operation of lines [5,6]. Previous studies have shown that the safe implementation of the line dynamic capacity-increase technology must be based on accurate DTR prediction to avoid potential line overheating risks [7].

Early DTR prediction studies relied primarily on the combination of physical models and weather forecasts [8,9]. However, the parameters of physical models often need to be determined based on manual experience, and the spatial scale of weather forecasts is excessively large, leading to significant errors compared with the high-precision micrometeorological data required for line capacity increase. In response, subsequent researchers began to propose utilizing the existing micrometeorological monitoring devices installed on transmission lines and adopting statistical models driven by real meteorological data—such as multiple linear regression [10] and least squares regression [11]—to predict DTR. Although this method can avoid issues such as parameter dependence and excessively large spatial scales, the linear superposition functions commonly used in statistical models struggle to characterize the nonlinear coupling effects of multiple meteorological factors on DTR, resulting in significant prediction errors. In recent years, artificial intelligence models with strong nonlinear characterization capabilities have gradually become a research focus in the field of DTR prediction. Early studies attempted to introduce the Elman Neural Network (Elman) [12], which demonstrated unique advantages in capturing short-term DTR temporal features by leveraging the dynamic recurrence mechanism of its hidden layer neurons, laying the foundation for subsequent temporal modeling. However, when processing long-time-span DTR data, this model has limited memory capacity, making it difficult to effectively capture DTR data features on longer time scales. In response, subsequent studies considered adopting models with stronger long-term temporal feature extraction capabilities for DTR prediction, such as the Temporal Convolutional Network (TCN) [13], Gated Recurrent Unit (GRU) [14],

and Bidirectional Long Short-Term Memory Network (BiLSTM) [15]. Although the aforementioned models have their own focuses in capturing features on different time scales, their common shortcomings persist in practical applications: single models are generally limited by their architectural characteristics, making it difficult to simultaneously and efficiently extract both short-term and long-term temporal features of DTR data within the same modeling framework. Specifically, the short-term recurrence advantages of EN and the long-term temporal feature extraction capabilities of models such as TCN form a mutually exclusive effect in existing network architectures, leading to imbalanced characterization of DTR multi-time-scale features and making it difficult to further improve prediction accuracy. Furthermore, changes in DTR dataset distribution caused by differences in operating environments of transmission lines in different regions also result in insufficient generalization ability of traditional single AI models with fixed structures.

Notably, relevant studies in the field of time-series prediction have provided important technical insights for DTR prediction. For instance, aiming at the nonlinearity and long-term dependence issues of runoff data, Xu et al. [16] proposed a TCN-CNN-LSTM hybrid model. By leveraging TCN to extract long-term temporal correlations, CNN to capture local multi-scale features, and LSTM to manage long-term memory, they constructed a multi-module collaborative modeling framework, whose core idea provides a valuable reference for the multi-scale feature capture of DTR data. However, this model does not involve adaptive hyperparameter optimization for data decomposition algorithms; moreover, it relies solely on single runoff time-series data and fails to consider the coupling effects of multiple meteorological factors (such as wind speed, temperature, and solar radiation) in DTR prediction, making it difficult to directly adapt to the complex formation mechanism of DTR data. In another representative study, Wang et al. [17] proposed a dual-channel parallel model based on Mamba and depth-gated attention layers. This model achieves efficient processing of long sequences via a selective state space mechanism, balancing prediction accuracy and computational efficiency. Nevertheless, the model focuses only on the single input of historical runoff sequences, which makes it unable to integrate the essential micrometeorological parameters for DTR prediction. Additionally, its complex architecture hinders it from meeting the near-real-time requirements of DTR prediction for large-scale power grids. These cross-domain studies have verified the effectiveness of the prediction framework combining hybrid models with hierarchical feature extraction in time-series prediction tasks. However, due to scenario differences, they have not yet addressed the core issues of DTR prediction and still require targeted optimizations for practical DTR forecasting scenarios.

In addition to the prediction limitations of single models with fixed network architectures, the non-stationary characteristics of DTR data itself are another key factor restricting the further improvement of prediction accuracy. To avoid the adverse impact of DTR data non-stationarity on prediction results, recent studies have generally tended to integrate decomposition algorithms—such as Empirical Mode Decomposition (EMD) [5] and Ensemble Empirical Mode Decomposition (EEMD) [6]—with the aforementioned single models to mitigate the adverse impact of the non-stationarity of raw data on the models. However, empirical decomposition algorithms do not impose constraints on the frequency of each DTR data component during decomposition, leading to frequency aliasing among components. In response, the latest studies in this field have attempted to use Variational Mode Decomposition (VMD) [7] to replace the aforementioned empirical decomposition algorithms for DTR data decomposition. By constructing and solving frequency constraint equations for each DTR component, VMD achieves adaptive decomposition of DTR components and can eliminate the frequency aliasing issue. Nevertheless, the decomposition performance of VMD highly depends on the reasonable configuration of two key hyperparameters: the number of decomposed components and the penalty factor. Previous studies primarily relied on manual experience to determine hyperparameters, lacking an intelligent optimization mechanism based on DTR data characteristics, which makes it impossible to consistently ensure optimal decomposition performance across different DTR datasets.

In view of this, this paper proposes an improved VMD-driven time-varying multi-model ensemble-based method for predicting the current-carrying capacity of transmission lines. First, the advantage of VMD in accurate frequency separation is utilized to decompose DTR data; meanwhile, the Sticky Mushroom Optimization Algorithm (SMA) is introduced

 

for adaptive iterative optimization of VMD hyperparameters. Second, a time-varying multi-model ensemble framework is established: the dynamic recurrence characteristic of Elman is used to capture the short-term temporal features of DTR data, and the multi-scale convolution structure of TCN is used to capture the long-term temporal features of DTR data, enabling modeling and prediction of DTR components with different characteristics. Finally, a dynamic weighting mechanism based on time-varying grey correlation coefficients is designed. By quantifying the real-time correlation between the outputs of the EN and TCN models and the actual values, weights are assigned to form adaptive ensemble prediction outcomes. Experimental results show that the prediction accuracy and generalization ability of the proposed method are significantly superior to those of traditional methods across different DTR datasets, providing a solution with greater theoretical depth and application value for the safe application of transmission line dynamic capacity-increase technology.

## 2. Introduction to line DTR and analysis of the limitations of current prediction methods

### DTR of transmission lines and analysis of its influencing factors

The physical essence of DTR for overhead transmission lines refers to the maximum current that a conductor can continuously carry while maintaining a stable temperature not exceeding the design threshold in the meteorological environment. Its value directly depends on the real-time heat exchange balance between the conductor and the surrounding meteorological environment [18]. From the perspective of heat transfer, the heat balance process of the conductor can be characterized by a heat balance equation: heat generation of the conductor mainly comes from Joule heat produced by resistance loss, while heat dissipation is achieved through the dynamic coupling of convective heat dissipation, radiative heat dissipation, and solar heat absorption. The dynamic nature of this heat exchange balance causes the DTR value to fluctuate in real time with changes in meteorological conditions, exhibiting complex multi-scale temporal features and non-stationary characteristics, which pose multiple challenges to high-precision prediction.

From the long-term time dimension, the variation patterns of solar radiation intensity and ambient temperature exhibit significant long-term periodic characteristics. Solar radiation is affected by astronomical factors such as the Earth's orbital revolution and the obliquity of the ecliptic, showing seasonal fluctuations with an annual cycle: in summer, the solar altitude angle is large and the sunshine duration is long, so the solar radiation heat absorbed by the conductor increases significantly, resulting in downward pressure on DTR during high-temperature periods; the opposite is true in winter, where the low-temperature environment creates favorable conditions for conductor heat dissipation, and DTR is generally at a relatively high level. The seasonal fluctuations of ambient temperature form a synergistic effect with solar radiation, and the two together constitute the dominant driving force for the long-term temporal changes of DTR, making the DTR data show obvious periodic fluctuation patterns on seasonal and annual scales.

In the short-term time dimension, the random volatility of wind speed and wind direction causes high-frequency disturbances to DTR. The magnitude of wind speed directly affects the convective heat transfer coefficient on the conductor surface: when the wind speed suddenly increases from 1 m/s to 5 m/s, the convective heat dissipation efficiency can increase by 3 times, promoting a significant rise in DTR in a short time; while changes in wind direction angle indirectly affect the convective heat exchange area by altering the relative angle between the air flow and the conductor axis. This high-frequency fluctuation of DTR caused by rapid changes in the wind field makes the DTR data fluctuate sharply on an hourly or even minute scale.

It is also worth noting that the coupling effect of the above four meteorological factors further intensifies the non-stationarity of DTR data. For example, at noon in summer, the superposition of strong solar radiation and high ambient temperature significantly increases the heat absorption rate of the conductor; if sudden gusts of wind occur at this time, the sudden rise in wind speed will break the original heat balance in a short time, leading to extreme non-stationary changes in DTR. On cloudy days in winter, the combination of low solar radiation and low ambient temperature makes conductor heat dissipation dominant, and the continuous change of wind direction angle at this time will cause small-amplitude

high-frequency oscillations of DTR. This multi-factor, multi-scale coupling effect makes DTR data exhibit obvious non-stationary characteristics, making it difficult for traditional stationary time series models to predict it accurately.

To sum up, the multi-scale temporal features and non-stationary characteristics of DTR data pose dual challenges to prediction models: on the one hand, the model architecture needs to have the ability to capture the long-term and short-term temporal features of DTR data respectively; on the other hand, the model needs to be as little affected by the non-stationarity of DTR data as possible. Therefore, constructing a prediction framework with both the ability to extract features of different time scales and a non-stationarity handling mechanism has become the core breakthrough for achieving high-precision DTR prediction.

## Defect mechanisms of current DTR prediction methods and key improvement areas

To address the non-stationary characteristics of DTR data, the VMD algorithm, which has emerged in recent years, has theoretically solved the mode aliasing issue commonly found in traditional empirical mode decomposition algorithms by introducing a variational constraint mechanism. This provides a new approach for the stationarization of non-stationary DTR data. However, it is noteworthy that the decomposition performance of the VMD algorithm is actually limited by the setting of two key parameters: the number of decomposed components and the penalty factor [19]. This characteristic imposes significant constraints on its practical application: even if the optimal parameter combination is determined through repeated trial-and-error in one DTR dataset, a significant drop in decomposition accuracy is likely to occur when the combination is directly transferred to datasets of different seasons.

To overcome this challenge, this paper proposes using the minimum envelope entropy of DTR data components as the objective function to quantify the impact of the number of decomposed components and the penalty factor on VMD decomposition performance. This converts the traditional experience-dependent VMD hyperparameter selection mode into an optimization problem solvable via quantitative indicators. Under this framework, the powerful global optimization capability of the SMA is further leveraged to perform intelligent iterative optimization within a preset hyperparameter space. By comparing the numerical differences of the fitness function during the iteration process, the optimal hyperparameter configuration of VMD is finally determined. By combining the quantitative evaluation of the fitness function with swarm intelligence optimization, this method not only eliminates the subjective bias of manual experience-based parameter selection and significantly mitigates the non-stationarity of DTR data but also greatly enhances the adaptability of the VMD algorithm across different DTR datasets.

Regarding the limitations of existing single models in multi-scale feature extraction, studies have found that various models exhibit feature preferences [20]: The Elman model, relying on its unique dynamic feedback mechanism, has a natural advantage in capturing high-frequency fluctuations caused by sudden wind speed changes. However, when processing the seasonal trends of solar radiation, it tends to suffer from prediction lag due to memory decay. In contrast, the TCN model, by virtue of the receptive field expansion capability of dilated convolution, can capture multi-scale periodic superposition effects but exhibits over-smoothing when handling non-periodic gusts. This limitation of network architecture—where a single model only adapts to one feature dimension of data—makes it difficult for model predictions to simultaneously ensure the long-term and short-term prediction accuracy of DTR data.

To break through this dilemma, this paper proposes a time-varying weight multi-model ensemble strategy. Guided by the core principle of "multi-time-scale modeling and dynamic collaboration," this strategy first constructs a dual-layer extraction system for the short-term and long-term temporal features of DTR data: On one hand, it utilizes the dynamic recurrence characteristic of the EN model to focus on capturing the short-term temporal features of DTR driven by high-frequency meteorological factors such as wind speed and wind direction. On the other hand, it relies on the multi-scale convolution structure of the TCN model to accurately characterize the long-term temporal features of DTR dominated by ambient temperature and solar radiation intensity. This enables refined modeling of the temporal features of DTR data across different scales. On this basis, the grey correlation coefficient is innovatively introduced to construct a

dynamic weight assignment mechanism. By quantitatively analyzing the correlation degree between the output results of the two aforementioned models and the actual values of real-time DTR data, adaptive adjustment of the contribution weights of different models in the prediction process is achieved. This mechanism not only allows the ensemble model to combine the advantages of single models in targeted modeling of different temporal features but also effectively enhances the generalization ability of the model across different DTR datasets from the architectural level.

## 3. Introduction to the principles of relevant algorithms

### VMD algorithm

The core of VMD in decomposing DTR data lies in determining the center frequency of each DTR data component by iteratively searching for the optimal solution of the variational model, thereby achieving effective separation of data components from low to high frequencies [21]. The separation process is as follows.

Firstly, To avoid frequency aliasing between DTR components, first construct a constrained variational equation with the objective of minimizing the sum of the bandwidths of each component:

$$\begin{cases} \min\limits_{\{u_k\},\{\omega_k\}} \left\{ \sum\limits_k \left\| h_t \left[ \left( \delta(t) + \frac{j}{\pi t} \right) * u_k(t) \right] e^{-j\omega_k t} \right\|_2^2 \right\} \\ \text{s.t.} \sum\limits_k u_k(t) = f(t) \end{cases} \tag{1}$$

Where: min{A}, s.t. B means minimizing A under the constraint of B; $h_t$ is the partial derivative operator; $\delta(t)$ is the Dirac delta function; j is the imaginary unit; π is pi (the circular constant); * denotes the convolution operator; $u_k(t)$ is the k-th component generated by DTR data decomposition; $w_k(t)$ is the frequency corresponding to the k-th component; $f(t)$ is the undecomposed DTR data.

Secondly, introduce the Lagrange multiplier $\lambda$ and penalty factor $\alpha$ to convert the constrained variational equation shown in Equation (2) into an unconstrained variational equation that is easier to solve:

$$\begin{aligned} L(\{u_k\}, \{\omega_k\}, \lambda) = & \\ \alpha \sum\limits_k \left\| h_t \left[ \left( \delta(t) + \frac{j}{\pi t} \right) * u_k(t) \right] e^{-j\omega_k t} \right\|_2^2 & \\ + \left\| f(t) - \sum\limits_k u_k(t) \right\|_2^2 + \left[ \lambda(t), f(t) - \sum\limits_k u_k(t) \right] \end{aligned} \tag{2}$$

Where: $\alpha$ is the penalty factor; k is the number of decomposed components of line DTR data; $\lambda$ is the Lagrange multiplier.

Thirdly, adopt the Alternating Direction Method of Multipliers to iteratively update $u_k$ and $w_k$ so as to find the optimal solution of Equation (3):

$$\hat{u}_k^{n+1}(\omega) = \frac{\hat{f}(\omega) - \sum\limits_{i \neq k} \hat{u}_i(\omega) + \frac{\hat{\lambda}(\omega)}{2}}{1 + 2\alpha(\omega - \omega_k)^2} \tag{3}$$

$$\begin{cases} \min\limits_{\{u_k\},\{\omega_k\}} \left\{ \sum\limits_k \left\| h_t \left[ \left( \delta(t) + \frac{j}{\pi t} \right) * u_k(t) \right] e^{-j\omega_k t} \right\|_2^2 \right\} \\ \text{s.t.} \sum\limits_k u_k(t) = f(t) \end{cases} \tag{4}$$

Where: n denotes the number of iterations; the Fourier transforms of $\hat{u}_k^{n+1}(\omega)$、$\hat{f}(\omega)$、$\hat{u}_k(\omega)$, and $\hat{\lambda}(\omega)$ correspond to $u_k^{n+1}(t)$、$f(t)$、$u_k(t)$ and $\lambda(t)$;

Fourth, iteratively update the above steps until the following iteration termination condition is met:

$$\sum_{k=1}^{K} \left( \frac{\| \hat{u}_k^{n+1}(\omega) - \hat{u}_k^n(\omega) \|_2^2}{\| \hat{u}_k^n(\omega) \|_2^2} \right) < \varepsilon \tag{5}$$

## Intelligent optimization algorithms

**3.1.1. SMA Algorithm.** To improve the DTR prediction accuracy, it is necessary to use an optimization algorithm for the hyperparameter optimization of VMD. As a new heuristic optimization method inspired by biological behaviors in nature, SMA demonstrates significant advantages in addressing parameter optimization problems, thanks to its characteristics such as strong adaptability and outstanding global optimization capability. It is particularly suitable for the parameter optimization task of the VMD algorithm. In view of this, this study employs SMA to optimize the hyperparameters of VMD when conducting line DTR prediction.

The specific mathematical model of the SMA Algorithm is as follows:

(1) Simulate slime mould individuals perceiving the presence of food and starting to move toward the food source. Among them, the position update formula for the i-th slime mould is:

$$X_i = \begin{cases} X_b(t) + \nu_b \cdot (W \cdot X_A(t) - X_B(t)), r < p \\ \nu_c \cdot X(t), r \geq p \end{cases} \tag{6}$$

Where: $X_A(t)$ and $X_B(t)$ denote two random individuals; $W$ is the weight coefficient; $X(t)$ denotes the current position; $X_b(t)$ denotes the optimal position; $v_b$ and $v_c$ are values in the range of [–a, a] and [–1, 1], respectively; $r$ is a random value in [0, 1]; $p$ is the control parameter.

Among them, the update formulas for parameters $p$, $a$ and the weight coefficient $W$ are as follows:

$$p = \tanh |S(i) - D_F| \tag{7}$$

$$a = \arctan h \left( -\frac{t}{T} + 1 \right) \tag{8}$$

$$W(I_{ndex}(i)) = \begin{cases} 1 + r_1 \lg \left( \frac{b_F - S(i)}{b_F - w_F} + 1 \right), 1 \leq i < \frac{N}{2} \\ 1 - r_1 \lg \left( \frac{b_F - S(i)}{b_F - w_F} + 1 \right), \frac{N}{2} \leq i \leq N \end{cases} \tag{9}$$

$$I_{ndex} = \text{sort}(S) \tag{10}$$

Where: S(i) is the fitness of the i-th slime mould individual, with i = 1, 2, …N; DF is the current optimal fitness; bF and wF represent the optimal fitness and worst fitness in the current iteration, respectively; t is the current iteration number; T is the maximum number of iterations; Index is the fitness sequence; r1 is a random number in [0, 1]; N is the number of slime mould individuals in the population.

(2) Once slime mould individuals approach the food, they will start to form a group surrounding the food to encircle the target. The formula for updating the position of slime mould is as follows:

$$X(t+1) = \begin{cases} r_{and} \cdot (U_B - L_B) + L_B, r_{and} < z \\ X_b(t) + \nu_b \cdot (W \cdot X_m(t) - X_n(t)), r_{and} < p \\ \nu_c \cdot X(t), r_{and} \geq p \end{cases} \tag{11}$$

Where: $U_B$ and $L_B$ represent the optimization range; $r_{and}$ denotes an arbitrary value within [0, 1]; $Xm(t)$ and $Xn(t)$ represent the positions of two randomly selected slime mould individuals; $Z$ is a user-defined parameter.

(3) When slime moulds seize food, individual slime moulds rely on propagating waves generated by biological oscillators to change the cytoplasmic flow in their veins. Through $v_b$, $v_c$, and $W$, it simulates the changes in the slime mould's vein width and the oscillation frequency of biological oscillators. The slime moulds continuously adjust their positions and select the optimal path and direction to obtain food. $W$ is an adaptive weight factor for positive and negative feedback, which simulates the change in oscillation frequency that continuously approaches 1 when the slime mould is in different food concentrations. $v_b$ oscillates randomly within [–a, a] and gradually approaches zero as the number of iterations increases. $v_c$ oscillates within [–1, 1] and eventually tends to zero; at this point, its mathematical model reaches the optimal state.

**3.1.2. Fitness Function.** When using the VMD algorithm to process DTR data, the selection of hyperparameters directly affects the decomposition performance, and hyperparameter optimization needs to be conducted through an optimization algorithm. The optimization process takes the fitness function value as the evaluation criterion, and envelope entropy is selected as the fitness function [22]. As a function that effectively characterizes data sparsity, envelope entropy exhibits a negative correlation between its value and the richness of effective feature information in DTR data: the more prominent the effective features are, the smaller the data envelope entropy is.

For each DTR component obtained by VMD decomposition under different parameter combinations, its envelope entropy value is calculated. The decomposition component with the smallest envelope entropy contains the richest effective feature information. Taking this component as the target item, the VMD parameters corresponding to this sequence are the optimal parameters.

The mathematical expression of envelope entropy is as follows:

$$e_i = \frac{h(i)}{\sum\limits_{i=1}^{K} h(i)}$$

(12)

$$E_e = -\sum_{i=1}^{K} e_i \lg e_i$$

(13)

Where: $h(i)$ is the envelope signal of $u(i)$ after Hilbert transform.

**3.1.3. Specific Process of VMD Hyperparameter Optimization via SMA.**

1. Definition of hyperparameter search space:

Clarify the ranges of two core hyperparameters (i.e., decomposition component number and penalty factor). The approximate boundaries of hyperparameters are determined according to the characteristics of DTR data and pre-experiments to eliminate invalid hyperparameter combinations.

2. Calculation of fitness function:

Based on information entropy theory, the envelope entropy of DTR components decomposed by VMD is adopted as the optimization objective to quantify the decomposition performance of VMD.

3. Generation of initial SMA population:

In line with population-based optimization theory, an initial SMA population covering the entire hyperparameter search space is generated, which ensures the global search capability of the SMA algorithm.

 

4.  Iterative update of SMA individual positions:

Following the biological mechanism of SMA, the positions of the population are updated via mathematical models to balance global exploration and local exploitation capabilities of the algorithm.

5.  Physical constraint verification and fitness recalculation of updated parameters:

The parameters that violate the physical constraints of VMD are corrected to ensure the validity of hyperparameters, thereby avoiding invalid computation in subsequent decomposition processes.

6.  Convergence criterion determination and output of optimal hyperparameters:

Based on numerical optimization theory, the optimization termination condition is defined according to the preset convergence criterion, which achieves a trade-off between optimization accuracy and engineering real-time performance, and finally outputs the optimal VMD hyperparameter combination.

## Elman model

The applicability of Elman in DTR prediction stems from its unique recursive structure, which enables highly efficient modeling of the short-term temporal features of data. As a sequence with strong temporal dependence, the correlation between the current state and historical states of DTR data can be categorized into short-term and long-term temporal features. TCN can address the issue of capturing long-term temporal features through dilated convolution, but it has limitations in capturing short-term data features. In response to this, Elman constructs a dynamic memory mechanism by introducing a context layer, directly feeding the hidden layer state at the previous time step back to the input layer to form a closed-loop recursive structure—this accurately captures the short-term correlations between adjacent time steps in DTR data (for detailed mathematical principles, see Reference [23]). In addition, the structure of the Elman model only consists of an input layer, a hidden layer, and a context layer. Compared with TCN, it has significantly fewer parameters, which not only improves training efficiency but also provides a lightweight supplement to the ensemble model.

To sum up, Elman effectively compensates for TCN's shortcomings in capturing the short-term temporal features of DTR and in computational efficiency. Combining Elman with TCN to construct an ensemble model can give full play to their respective advantages, realizing the collaborative modeling of the long-term and short-term temporal features of DTR data.

## TCN model

TCN is a neural network model that integrates dilated causal convolution modules and residual connection modules, and its network architecture is shown in Fig 1

TCN ensures that the model input only contains historical meteorological data through its causal convolution structure, thereby avoiding prediction bias caused by future data leakage. On this basis, its exponential dilated convolution structure further enhances the model's ability to extract long-term temporal features of DTR. Specifically, as the dilation factor grows exponentially according to the rule of $d = 2i$, a 3-layer network can form a receptive field of 29 time steps. This directly corresponds to the time scales of solar radiation intensity and ambient temperature in the process of conductor thermal balance, ensuring the model can capture the key moments in the evolution of long-term temporal features. Given that the prediction principle of the TCN model is well-established, it will not be repeated here; the detailed mathematical derivation process can be found in Reference [24].

## Time-varying grey relational coefficient method

The key to an ensemble model lies in the dynamic optimization of weight allocation. Traditional static fixed-weight methods cannot adapt to the dynamic changes in the performance of individual models during DTR prediction, which restricts the improvement of the ensemble model's prediction accuracy [24]. To address this, this paper proposes a dynamic weight

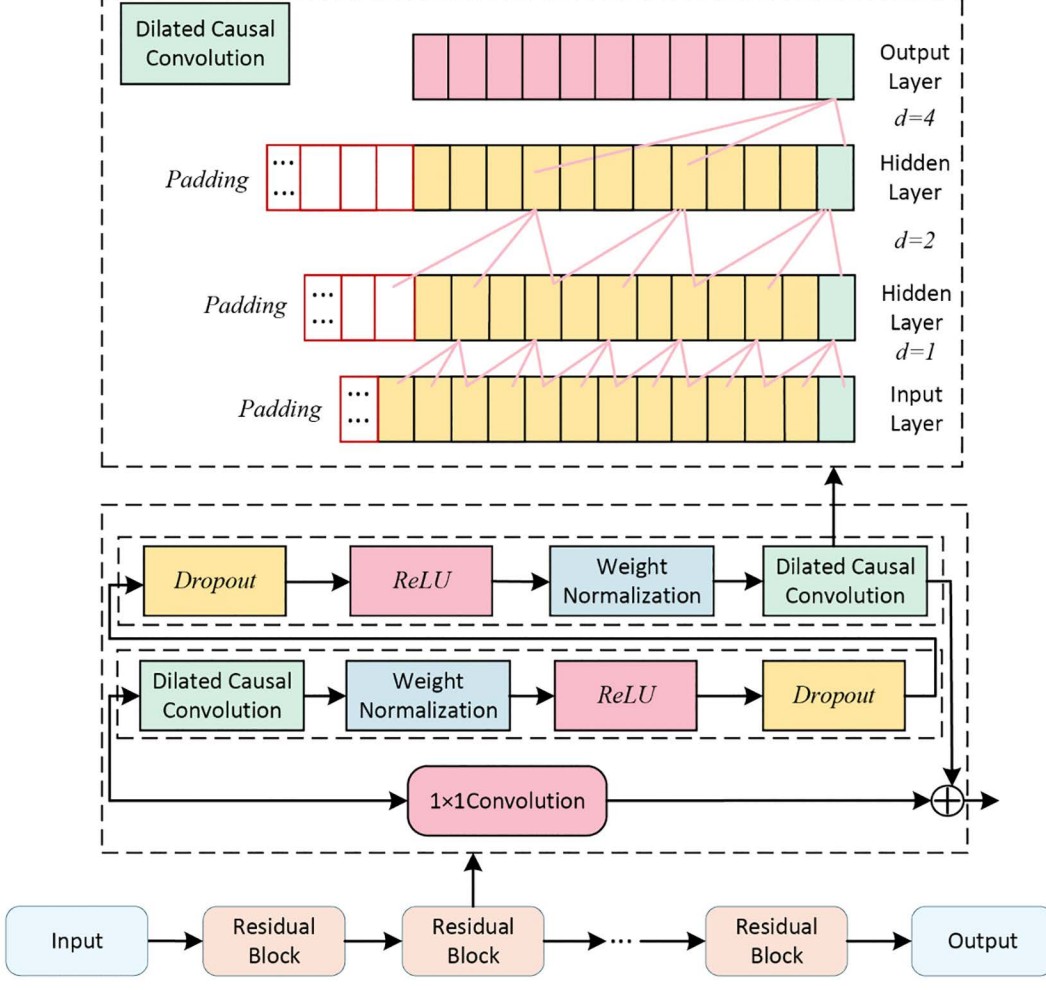

**Fig 1. TCN Model.**

allocation mechanism based on the Time-varying Grey Relational Coefficient Method. Its core lies in dynamically evaluating the matching degree between the output of each model and the actual DTR according to the real-time states of long-term and short-term features in DTR data. This enables the automatic switching of the dominant model based on real-time meteorological features, avoiding the situation where the performance shortcomings of a single model in non-advantageous scenarios affect the overall DTR prediction performance.

In the actual prediction process, since the current value is unknown, it is impossible to directly calculate the time-varying weights based on the current value. Therefore, we consider allocating weights based on historical prediction performance.

(1) Construct an analysis matrix of the actual DTR values and the predicted values of each individual model:

$$(\boldsymbol{X_t, Y_t}) = \begin{bmatrix} x_1(1) & \cdots & x_i(1) & \cdots & x_u(1) & y(1) \\ \vdots & \ddots & \vdots & \ddots & \vdots & \vdots \\ x_1(t) & \cdots & x_i(t) & \cdots & x_u(t) & y(t) \\ \vdots & \ddots & \vdots & \ddots & \vdots & \vdots \\ x_1(T) & \cdots & x_i(T) & \cdots & x_u(T) & y(T) \end{bmatrix} \tag{14}$$

Where: $X_1$, $X_2$, …, $X_u$ are the sequences of historical predicted DTR values of each individual model; 1, 2, $t$, $T$ represent the range of historical data length; $y$ is the sequence of actual DTR values.

(2) Further perform normalization on the above-mentioned matrix to form an initial value matrix.

(3) Perform element-wise difference calculation on the elements in the initial value matrix:

$$\boldsymbol{\omega}_i(t) = \begin{bmatrix} \omega_1(1) & \cdots & \omega_i(1) & \cdots & \omega_m(1) \\ \vdots & \ddots & \vdots & \ddots & \vdots \\ \omega_1(t) & \cdots & \omega_i(t) & \cdots & \omega_m(t) \\ \vdots & \ddots & \vdots & \ddots & \vdots \\ \omega_1(T) & \cdots & \omega_i(T) & \cdots & \omega_m(T) \end{bmatrix} \tag{15}$$

(4) Select the maximum and minimum values from Equation (15):

$$\begin{cases} \omega_{\max} = \max_i \left( \max_t \omega_i(t) \right) \\ \omega_{\min} = \min_i \left( \min_t \omega_i(t) \right) \end{cases} \tag{16}$$

(5) Calculate the relational coefficients between the predicted values of each individual model and the actual values to form a relational coefficient matrix:

$$\lambda_i(t) = \frac{\omega_{\min} + \rho\omega_{\max}}{\omega_i(t) + \rho\omega_{\max}} \tag{17}$$

$$\lambda_i(t) = \begin{bmatrix} \lambda_1(1) & \cdots & \lambda_i(1) & \cdots & \lambda_m(1) \\ \vdots & \ddots & \vdots & \ddots & \vdots \\ \lambda_1(t) & \cdots & \lambda_i(t) & \cdots & \lambda_m(t) \\ \vdots & \ddots & \vdots & \ddots & \vdots \\ \lambda_1(T) & \cdots & \lambda_i(T) & \cdots & \lambda_m(T) \end{bmatrix} \tag{18}$$

Where: $\rho$ is the distinguishing coefficient, and this paper sets it to 0.5.

(6) Calculate the mean of each column in the relational coefficient matrix to obtain the grey relational degree between each individual model and the actual DTR values:

$$r_i = \frac{1}{n}\sum_{t=1}^{T} \lambda_i(t) \tag{19}$$

## Comprehensive evaluation indicator system

To comprehensively evaluate the performance of the model proposed in this paper in DTR data prediction, it is necessary to select targeted evaluation indicators by considering the non-stationarity and multi-scale characteristics of DTR data, as well as the practical requirements for prediction accuracy in engineering applications. The selection logic is as follows:

First, the root mean square error (RMSE) is significantly sensitive to large deviations in prediction results. In DTR data, factors such as sudden changes in wind speed are likely to cause short-term violent fluctuations. If such large errors are not effectively captured, it will lead to potential safety hazards in the assessment of transmission line current-carrying

capacity. Therefore, RMSE is selected to focus on measuring the prediction accuracy of the model for such key abnormal fluctuations, providing a quantitative basis for engineering safety.

Second, the prominent advantage of the mean absolute percentage error (MAPE) is that it is not affected by the magnitude differences of the original data and is more sensitive to changes in small errors. The value range of DTR data may fluctuate significantly due to differences in environment and time periods. MAPE can accurately reflect the model's ability to capture subtle fluctuations while eliminating the interference of magnitudes, making it suitable for evaluating the stability of the model under different environments or time periods.

Third, the mean absolute error (MAE) can intuitively reflect the average absolute deviation between predicted values and actual values, and its linear nature enables it to objectively reflect the overall prediction error level of DTR data [25]. The introduction of MAE can measure the comprehensive prediction effect of the model from an overall perspective, avoiding the excessive amplification of local errors by a single indicator.

Finally, the coefficient of determination ($R^2$) is used to characterize the degree of fitting of the prediction model to the inherent laws of DTR data. DTR data is affected by the coupling of multiple variables such as meteorological factors, and its variation laws are highly complex. The closer the value of $R^2$ is to 1, the more fully the model captures the potential correlations in the data, which can provide a basis for subsequent model optimization [26].

In summary, four indicators—RMSE, MAPE, MAE, and $R^2$—are selected to comprehensively evaluate the performance of the prediction model from four dimensions: sensitivity to large errors, adaptability to magnitudes, overall deviation level, and degree of law fitting.

$$\text{RSME} = \sqrt{\frac{\sum_{i=1}^{n} (y_i - \bar{y}_i)^2}{n}} \tag{20}$$

$$\text{MAPE} = \frac{1}{n} \sum_{i=1}^{n} \left| \frac{y_i - \bar{y}_i}{y_i} \right| \times 100\% \tag{21}$$

$$R^2 = 1 - \frac{\sum_{i=1}^{n} (y_i - \hat{y}_i)^2}{\sum_{i=1}^{n} (y_i - \bar{y}_i)^2} \tag{22}$$

$$\text{MAE} = \frac{1}{n} \sum_{i=1}^{n} |y_i - \bar{y}_i| \tag{23}$$

where: $n$ denotes the total number of samples of DTR data; $y_i$ denotes the actual values of line DTR data; $\bar{y}_i$ denotes the predicted values of DTR data.

## 4. Prediction framework

The line DTR prediction framework for the transmission line current-carrying capacity prediction method with time-varying multi-model integration follows the hierarchical processing logic of "DTR Data Quality Control - DTR Feature Processing - Multi-Model Collaborative Prediction", as shown in Fig 2.

(1) Data Quality Control: Collect transmission line DTR data and synchronized meteorological data. Use the cubic spline interpolation method for missing value imputation to ensure data continuity.

(2) Data Standardization: Based on Step (1), perform normalization on the entire dataset through min-max normalization to eliminate the impact of dimension differences on model training.

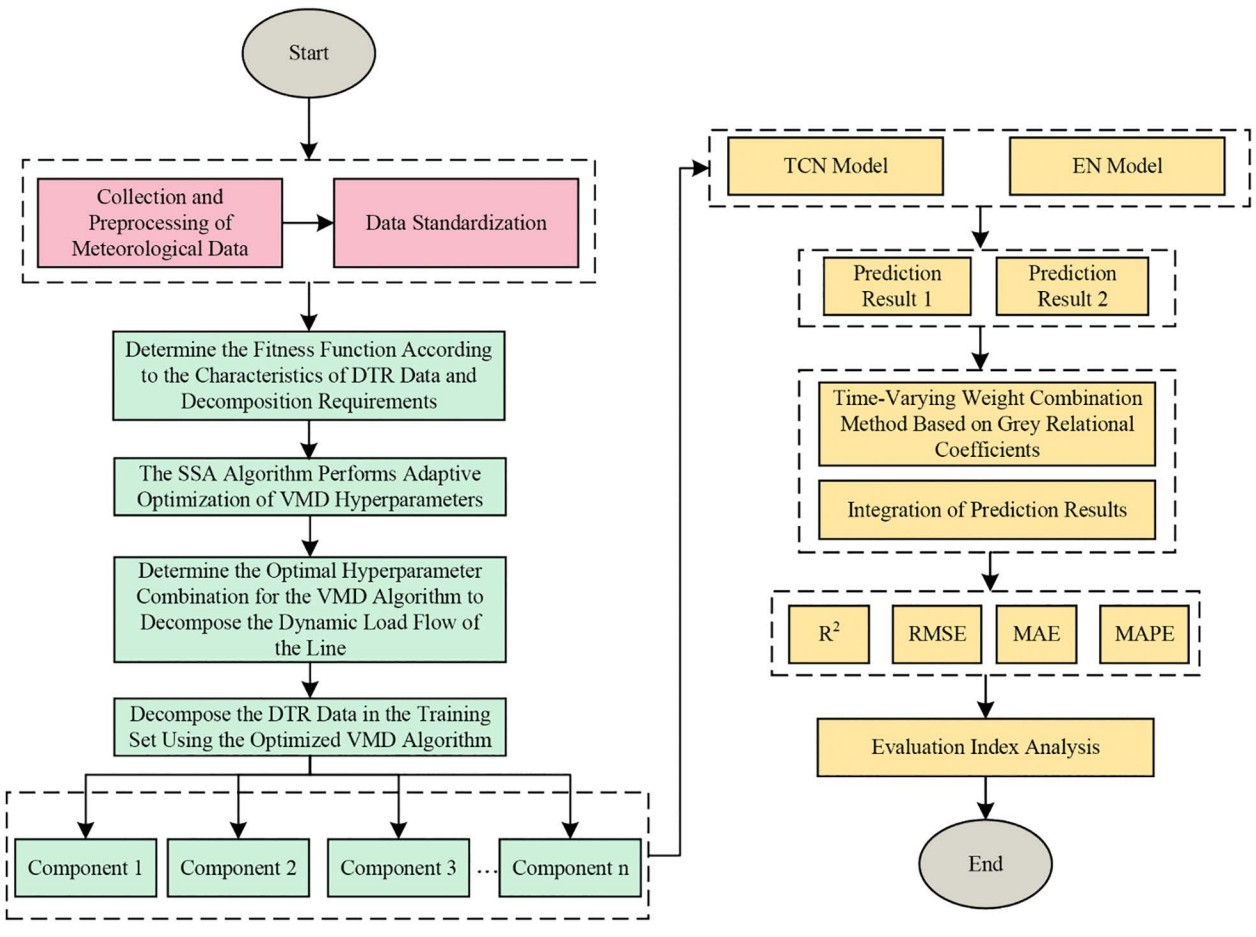

**Fig 2. DTR data forecasting process.**

(3) SMA-Driven VMD Hyperparameter Optimization: Take the minimization of the envelope entropy of decomposed DTR components as the objective function to construct a data-driven hyperparameter optimization model. Conduct iterative optimization within the preset parameter space using the SMA algorithm to obtain the optimal hyperparameter combination.

(4) VMD Decomposition of Non-Stationary DTR Data: Use the optimized VMD algorithm to decompose the normalized DTR data, adaptively separating the non-stationary data into a series of locally stationary components with different frequencies. This provides stationary input data for subsequent multi-scale modeling.

(5) Multi-Model Multi-Scale Feature Prediction: Utilize the dynamic recursive mechanism of the EN model to capture short-term DTR temporal features caused by sudden changes in wind speed and direction; use the TCN model to process long-term DTR temporal features caused by periodic changes in solar radiation intensity and ambient temperature.

(6) Time-Varying Weight Dynamic Integration Prediction: Incorporate the grey relational coefficient to construct a real-time weight allocation mechanism. By calculating the dynamic relational degree between the output of each individual

model and the actual values, adaptively adjust the prediction contribution to avoid feature extraction bias of a single architecture.

(7) Multi-Indicator Performance Evaluation: Use indicators such as RMSE, MAPE, $R^2$, and MAE to compare the prediction performance of the proposed method and traditional methods from multiple dimensions.

## 5. Case verification

### Description of case data sources

In recent years, Zhuhai City, China, has experienced rapid economic development, with the annual average growth rate of its maximum electricity load remaining among the top in the country. The surge in electricity demand has caused the power transmission capacity of some transmission lines to approach the current-carrying capacity limit, making the power transmission bottleneck increasingly prominent. There is an urgent need to tap the transmission potential of lines through dynamic capacity-increasing technology.

As a typical city in the southern subtropical monsoon climate zone, Zhuhai's meteorological elements exhibit unique complexity: the alternation of monsoons brings about significant seasonal abrupt changes, while the urbanization process triggers micro-environmental disturbances, resulting in obvious differences in the spatiotemporal distribution of parameters such as wind speed and temperature. Such complex climatic conditions pose a stringent test for the accuracy of DTR prediction models and provide a highly representative scenario for verifying model performance.

Therefore, this paper selects an overhead transmission line of 110kV voltage class in Zhuhai City as the research object. This line has a total length of 12.8 km and passes through diverse landforms including industrial areas and residential areas. It adopts LGJ-400/20 type steel-cored aluminum strands, with specific parameters shown in Table 1.

Since the meteorological characteristics of the environment where the line is located change significantly with seasons, and the model's performance in different seasons directly determines its practical application value, the verification of the model's accuracy and generalization ability must address the abrupt changes in meteorological characteristics caused by seasonal transitions. In view of this, this study divides the data collection window by seasonal boundaries, focusing on the capacity-increasing demand of transmission lines during summer and winter to maximize the test of the model's generalization ability. The specific time periods are divided as follows: (1) Dataset 1: July 3 to August 25, 2023, which falls in the midsummer, characterized by the interweaving of high temperatures and strong sunlight; (2) Dataset 2: January 8 to March 1, 2024, covering the period from midwinter to early spring, with prominent dominant characteristics of radiative heat dissipation in low-temperature environments. All data are recorded at 15-minute intervals, with the complete raw datasets available in S1 File.

It should also be noted that considering the differences in the spatial distribution characteristics of various meteorological parameters, to ensure that the collected data can truly reflect the environment around the line, the data acquisition adopts the spatial heterogeneity adaptation principle [27]: environmental temperature and solar radiation, which are regionally macroscopic, are monitored by national reference meteorological stations within 3 kilometers of the line. These stations have undergone metrological certification to ensure the regional representativeness of the data; while wind

**Table 1. Line parameters.**

| Parameter Type | Parameter Value |
| --- | --- |
| Conductor Model | LGJ-400/20 |
| Conductor Outer Diameter | 26.91mm |
| Conductor Maximum Allowable Temperature | 80°C |
| Conductor DC Resistance | 0.7104Ω/km |

parameters, which have strong microscale characteristics, are significantly affected by terrain turbulence at the tower locations. Therefore, integrated wind measurement devices are installed at some key towers along the line to directly obtain real wind field data around the conductors.

## VMD Decomposition experiment

Given that the DTR data during the study period is driven by complex meteorological processes, its time series exhibits significant non-stationarity characteristics. Therefore, it is necessary to uncover its intrinsic composition through an effective decomposition method. For this purpose, this study adopts a hyperparameter-optimized VMD algorithm, which separates the multi-scale non-stationary components contained in the original DTR sequence through an adaptive decomposition process, thereby providing stationary components for subsequent predictive modeling. The individual DTR components obtained from the decomposition are shown in Fig 3.

VMD decomposes the original DTR data into 9 locally stationary components with different central frequencies. Among them, the first component mainly reflects the overall variation trend of DTR data, representing the slow variation characteristics of DTR data on a long-time scale. This component is usually closely associated with long-term meteorological factors such as ambient temperature and solar radiation, featuring low frequency and slow variation. Separating the low-frequency trend component of DTR data through VMD helps subsequent prediction models better capture the slow variations of DTR data.

The remaining components respectively represent the fluctuation characteristics of DTR data in different frequency ranges. For example, Components 2–4 reflect the periodic variations of DTR data—such as DTR changes caused by

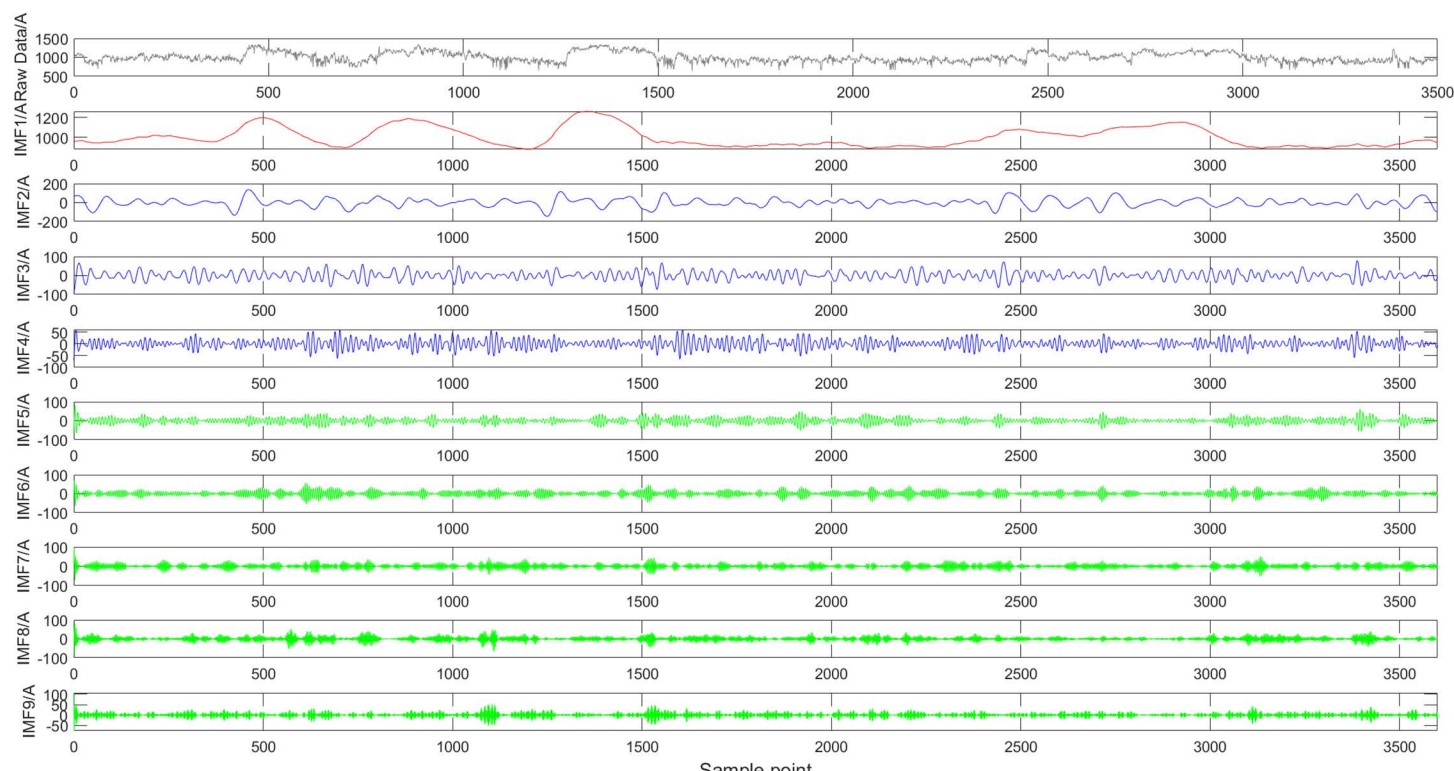

**Fig 3. Improved VMD decomposition of DTR data.** Legend: IMF1-IMF4 = intrinsic mode functions from improved VMD; Res = residual component; X-axis = time (h); Y-axis = DTR value (A).

diurnal periodic fluctuations of meteorological parameters (e.g., wind speed and temperature). Compared with the original DTR data, the non-stationarity of Components 2–4 has been significantly reduced, providing important intermediate-scale features for subsequent prediction models and contributing to the improvement of model prediction accuracy.

The rest of the components mainly reflect the random fluctuation characteristics of DTR data, such as rapid DTR changes caused by transient meteorological factors (e.g., gusts and short-term temperature fluctuations). These components frequently have higher frequencies and change more drastically than the trend and periodic components. Even though non-stationarity still exists in each high-frequency component, the amplitude of their non-stationarity has been greatly reduced, and the adverse impact on the overall prediction results has also been suppressed to a certain extent.

In summary, VMD decomposes complex non-stationary DTR data into multiple relatively stationary components, which significantly reduces the modeling difficulty of prediction models. Specifically, the separation of the trend component frees the model from the interference of short-term meteorological fluctuations; the extraction of periodic components helps the model capture the periodic characteristics of DTR; and the amplitude of non-stationarity in the remaining components is greatly reduced. Therefore, VMD provides more refined feature inputs for prediction models, which is conducive to improving prediction accuracy.

## VMD hyperparameter optimization experiment

Optimization algorithms can be divided into swarm intelligence optimization algorithms and non-swarm intelligence optimization algorithms. Swarm-based algorithms perform search through multi-agent cooperation and are suitable for global optimization; non-swarm-based algorithms rely on single-agent iteration and focus on local optimization. To comprehensively verify the advantages of SMA in VMD hyperparameter optimization, this study first compares SMA with non-swarm intelligence algorithms to examine its breakthroughs under the single-agent paradigm; then compares it with swarm intelligence algorithms to verify its competitiveness in swarm cooperation, thereby comprehensively evaluating its performance.

To verify the performance superiority of SMA in optimizing VMD hyperparameters, this study first selects two classic non-swarm intelligence optimization algorithms in the field of parameter optimization as comparison benchmarks. The comparison algorithms are the Simulated Annealing Algorithm (SA) and the Tabu Search Algorithm (TS). The iteration curves of each algorithm are shown in Fig 4, and the complete iteration curve data is provided in S2 File.

In terms of optimization speed, the value of SMA shows an obvious downward trend at the early stage of iteration: it drops from the initial 8.52 to 8.39 and basically remains stable. This indicates that SMA can quickly reduce the fitness function value to a relatively optimal level with a small number of iterations, exhibiting fast optimization speed. By contrast, SA has a high value in the early iteration stage: it remains at 8.95 for the first two iterations and only starts to drop to 8.93 at the 4th iteration. Moreover, the subsequent decline process is slow and accompanied by fluctuations, which suggests that SA has a slow optimization process—likely due to continuous exploration of different regions, which affects its convergence speed. In addition, TS also has a high value in the early iteration stage: it stays at 8.87 for the first three iterations and further drops to 8.80 at the 6th iteration before stabilizing. Although TS can stabilize relatively quickly, SMA reduces the fitness function value to a lower level within fewer iterations (in terms of both the magnitude of value reduction and the number of iterations to reach stability), resulting in relatively faster optimization speed.

In terms of optimization accuracy, SMA stabilizes at a relatively low fitness function value of 8.39 in the later stage of iteration. A lower fitness function value usually means that parameter combinations closer to the optimal solution are found during the optimization process. When SA completes 20 iterations, its fitness function value is around 8.86, which is significantly higher than SMA's stable value. This indicates that SA may not find a solution as optimal as SMA's, exhibiting relatively lower optimization accuracy. The stable fitness function value of TS is 8.80, which is also higher than SMA's stable value of 8.39. Judging from the final stable fitness function value, SMA can find a more optimal solution and outperforms SA and TS in terms of optimization accuracy.

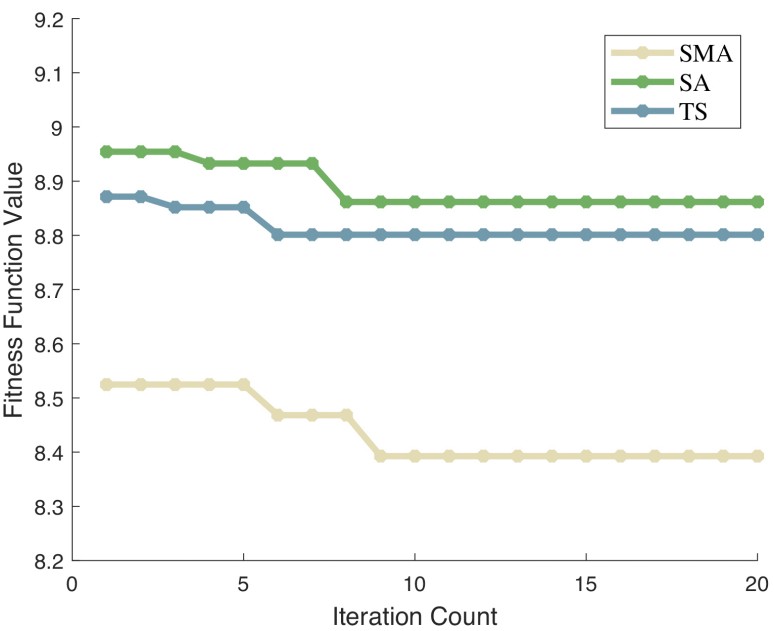

**Fig 4. Optimization performance comparison results.**

To further investigate the superior optimization performance of SMA, three representative swarm intelligence optimization algorithms are selected as a comparative reference system, namely Particle Swarm Optimization (PSO), Whale Optimization Algorithm (WOA), Grey Wolf Optimizer (GWO) and Arctic Puffin Optimization Algorithm(APO). By establishing a unified experimental environment and evaluation criteria, the iterative optimization process of each algorithm in the VMD hyperparameter optimization task is tracked and recorded. The curves of the objective function values changing with the number of iterations for each algorithm during the optimization process are shown in Fig 5.

In terms of optimization speed, the SMA algorithm exhibited a notable downward trend in fitness values at the initial stage of iteration, dropping from the initial value of 8.52 to 8.39 and stabilizing basically, demonstrating a fast optimization speed. By contrast, the PSO algorithm presented a relatively high fitness value (8.89) in the first iteration; although the value decreased gradually with the progress of iteration, the overall decline process was rather gentle, with a slower descending rate than that of SMA. The WOA and GWO algorithms showed similar value-decreasing trends. However, WOA did not exhibit an obvious decline until the 5th iteration, with its fitness value dropping to 8.62 at that point, while GWO presented an even slower variation in fitness values in the early iteration stage. Both algorithms required a significantly larger number of iterations than SMA to reach a stable state close to SMA's optimal fitness value. In addition, although APO had a lower initial fitness value than PSO, WOA, and GWO and started to decrease from the 2nd iteration, its descending rate was slow, only dropping to 8.4716 by the 7th iteration. It took approximately 18 iterations for APO to finally stabilize at 8.4168, which was still more than the number of iterations consumed by SMA. Therefore, SMA has a distinct advantage in optimization speed over the aforementioned population-based algorithms.

Regarding optimization accuracy, the SMA algorithm stabilized at a relatively low fitness function value of 8.39 in the later iteration stage. Generally, a lower fitness function value indicates that a parameter combination closer to the optimal solution has been obtained in the optimization process. At the end of 20 iterations, the fitness function value of PSO was around 8.79, which was higher than the stable value of SMA, suggesting that PSO failed to find a solution as optimal as that of SMA and thus had low optimization accuracy. The stable fitness values of WOA and GWO were approximately

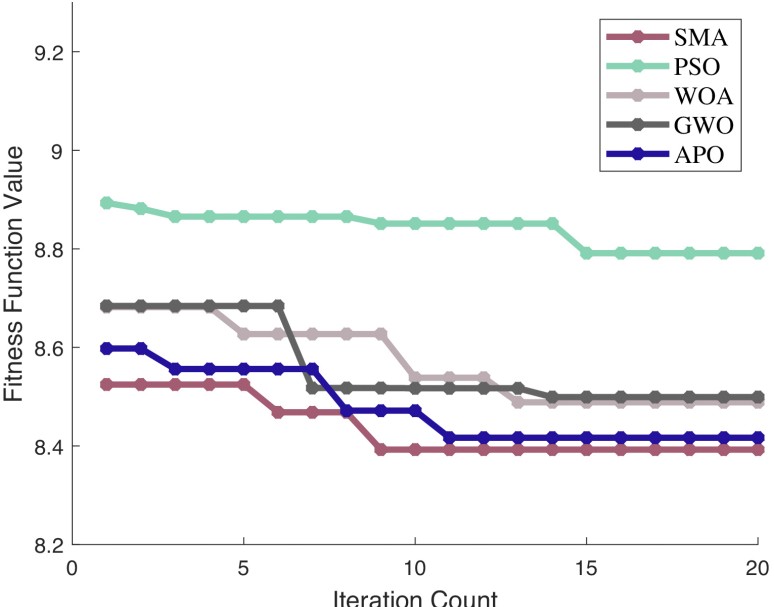

**Fig 5. Optimization performance of different algorithms.** Legend: Algorithms compared: SMA (Smile Mould Algorithm). PSO (Particle Swarm Optimization), GA (Genetic Algorithm); Y-axis = Fitness Function Value; X-axis = Iteration Count.

8.48 and 8.49, respectively; although these values were close to each other, they were still higher than 8.39 of SMA. The stable fitness value of APO was 8.4168, which was superior to those of WOA (8.4887), GWO (8.4992), and PSO (8.7912), yet it was still 0.0241 higher than that of SMA, showing a significant accuracy gap. Judging from the final stable fitness function values, SMA is capable of finding a more optimal solution and outperforms PSO, WOA, GWO, and APO in terms of optimization accuracy.

Based on a comprehensive comparative analysis of both optimization speed and optimization accuracy, the SMA algorithm demonstrates distinct advantages in VMD hyperparameter optimization compared with both non-swarm-based and swarm-based optimization algorithms.

**Comparative experiment**

To verify the significant advantages of the model proposed in this study in terms of prediction accuracy, five typical traditional models widely applied in the field of DTR prediction are selected as comparison benchmarks. The included comparative model system covers the Transformer model, LSSVM model, GRU model, TCN model, and BP model. These five comparative models can be divided into two categories based on model complexity and feature learning capabilities, with the classification reasons as follows:

The first category encompasses deep time-series modeling models, including Transformer, GRU, TCN, and LSTM models. All models in this category are built on deep learning architectures and possess robust capabilities for extracting time-series features, with their respective core mechanisms as follows:

The Transformer captures global dependencies through its self-attention mechanism, enabling the quantification of correlation weights across different time steps of sequential data.

The GRU optimizes temporal memory via its gated structure, balancing computational complexity while effectively retaining short-term fluctuation information of time-series data.

The TCN excavates multi-scale periodic features by virtue of dilated convolution, whose exponentially expanded receptive field facilitates the identification of long-term evolutionary patterns in sequential signals.

The LSTM dynamically filters, updates, and leverages its internal memory state through a specialized gating mechanism, thereby capturing and utilizing long-term dependencies within sequence data to support predictive tasks.

The second category is traditional and basic learning models, covering LSSVM and BP. As a classic machine learning method, LSSVM handles nonlinear relationships through kernel functions; the BP model, as a basic neural network, realizes basic nonlinear fitting through backpropagation and is a common tool for early temporal prediction.

Due to space constraints of the paper, Fig 6 and 7 show visual comparisons of the prediction results of each model in intervals with significant fluctuation characteristics in Dataset 1. Fig 8 presents quantitative results based on multiple evaluation indicators, with the complete comparative data of all models available in S3 File.

The model proposed in this study demonstrates significant advantages across all four evaluation indicators. Compared with the second-best Transformer model, it achieves a 21.02% increase in $R^2$, a 64.89% decrease in RMSE, and reductions of 64.85% in MAE and 64.94% in MAPE, respectively. This advantage stems from the model's multi-scale temporal feature capture mechanism and its ability to mitigate non-stationarity. Specifically, the model accurately captures short-term DTR fluctuations through the dynamic recursive structure of Elman, while the causal convolution of TCN collaboratively extracts long-term dependencies—effectively compensating for the limitation of Transformer, which relies solely on the self-attention mechanism. Furthermore, the model effectively reduces the non-stationary characteristics of DTR data through the integration of the SMA-VMD algorithm, significantly enhancing its prediction performance.

Horizontal comparison with other models reveals that although TCN and GRU possess temporal modeling capabilities, their RMSE values reach 52.9109 and 60.7323, respectively—exhibiting significantly lower prediction performance compared to the proposed model. This phenomenon reflects that the convolutional structure of TCN and the gating units of GRU provide insufficient handling of the non-stationarity in DTR data, leading to significantly increased errors during

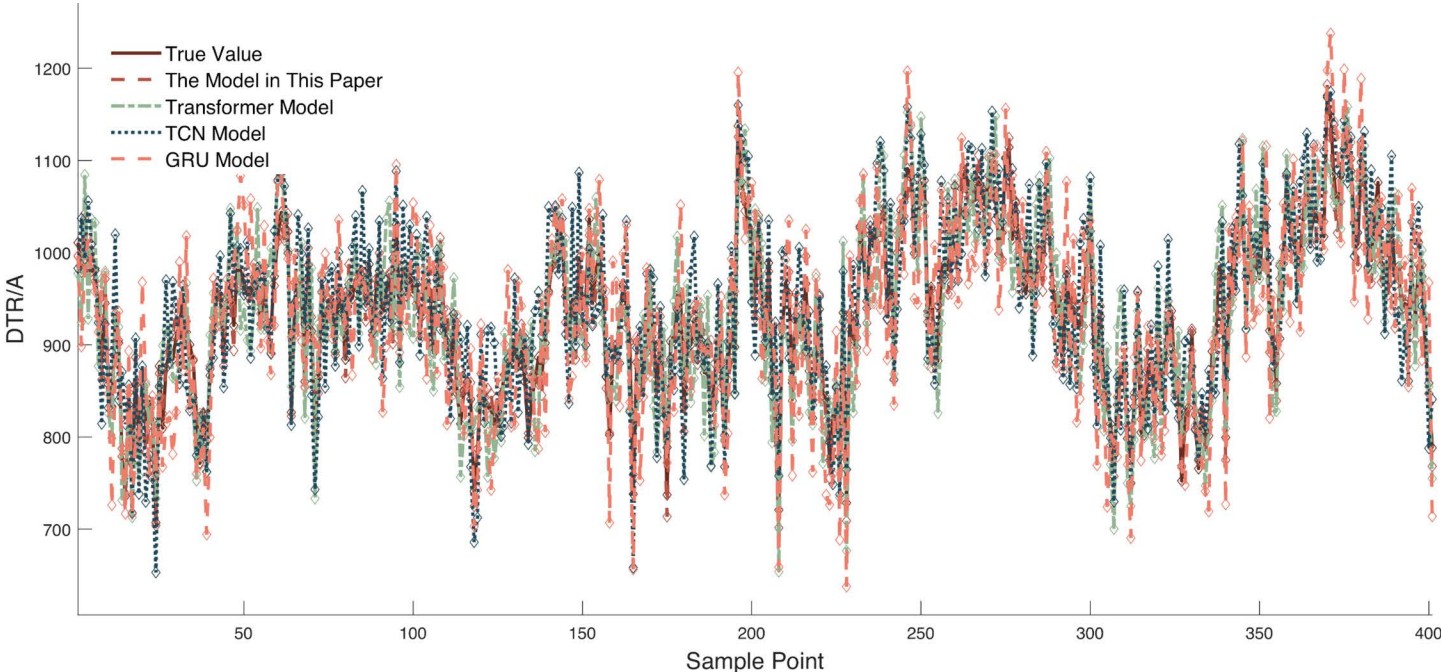

**Fig 6. Comparison of prediction results between the proposed model and deep time series modeling. model.** Legend: "proposed model"＝improved VMD-time-varying multi-model ensemble; deep time-series: LSTM, GRU, Transformer; X-axis＝Sample point; Y-axis＝predicted DTR (A).

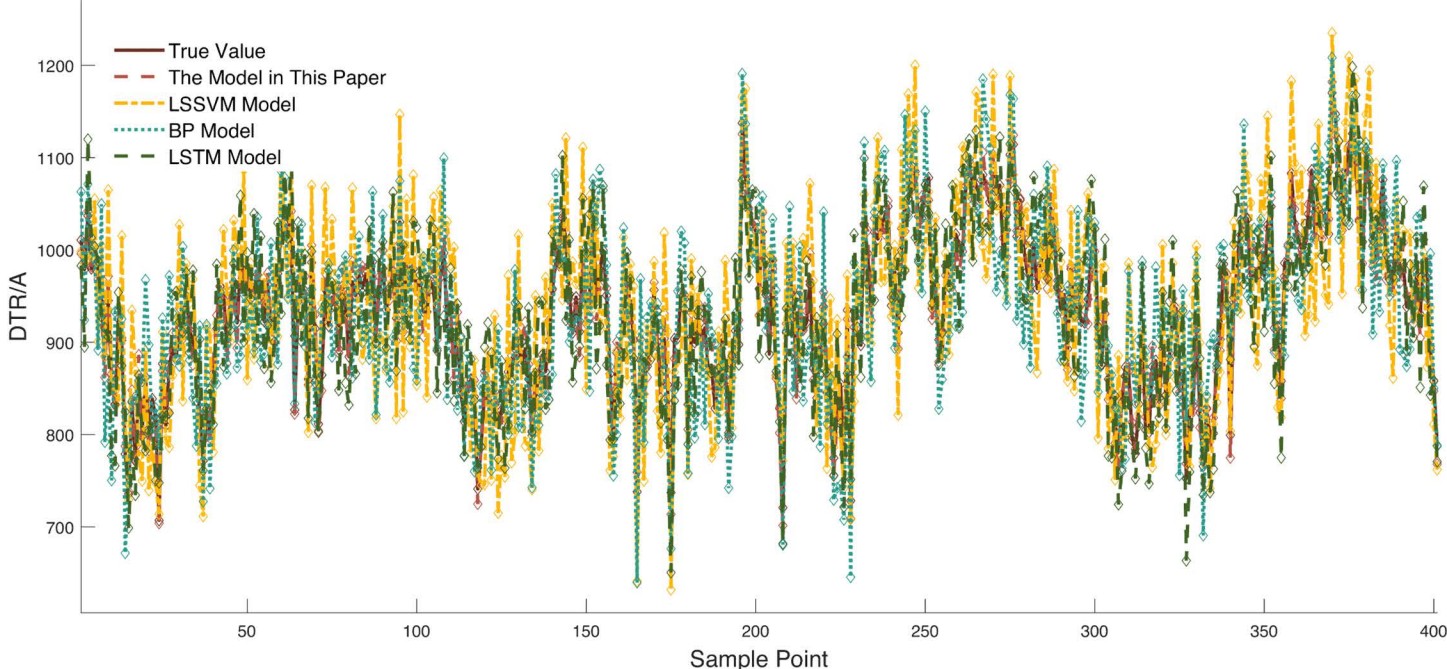

**Fig 7. Comparison of prediction results between the proposed model and traditional learning models.** Legend: "proposed model" = improved VMD-time-varying multi-model ensemble; Other models: BP (Back Propagation), LSSWM (Least· Squares Support Vector Machine); X-axis = Sample point; Y-axis = predicted DTR (A).

periods of rapid meteorological fluctuations. In contrast, the proposed model incorporates the SMA-VMD coupled algorithm to decompose DTR data, which greatly mitigates the non-stationarity of DTR data and thus enables the model to achieve significant improvements across multiple indicators.

As a classical model, the LSTM network is equipped with strong capabilities to capture long-term temporal dependencies by virtue of its gated structures such as the forget gate and input gate. However, it also fails to incorporate dedicated optimization modules targeting the non-stationary characteristics of DTR data, resulting in prediction performance that is still far inferior to that of the proposed model in this paper: the coefficient of determination $R^2$ of LSTM is only 67.91%, which is 28.0% lower than that of the proposed model; its RMSE reaches as high as 59.1816, representing a 69.2% increase relative to the proposed model. This result further confirms that relying solely on temporal modeling capabilities is insufficient to adapt to the non-stationary fluctuation characteristics of DTR data, and it is imperative to introduce targeted mechanisms for mitigating non-stationary features.

However, the LSSVM and BP models lack both temporal memory units and non-stationary feature handling mechanisms, making it difficult for them to accurately predict DTR data—whether for long-term trends or short-term fluctuations. It is evident that the $R^2$ of the LSSVM model is only 59.84%, a 36.07% decrease compared to the proposed model; the MAPE of the BP model reaches as high as 5.83%, an increase of 4.21% compared to the proposed model. These data effectively confirm the limitations of traditional models in modeling the temporal and non-stationary relationships of DTR data.

### Ablation experiment

To systematically verify the effectiveness of each core algorithm in the proposed model, the model itself is used as a baseline reference to construct a differentiated comparative model system, which includes three types: (1) The

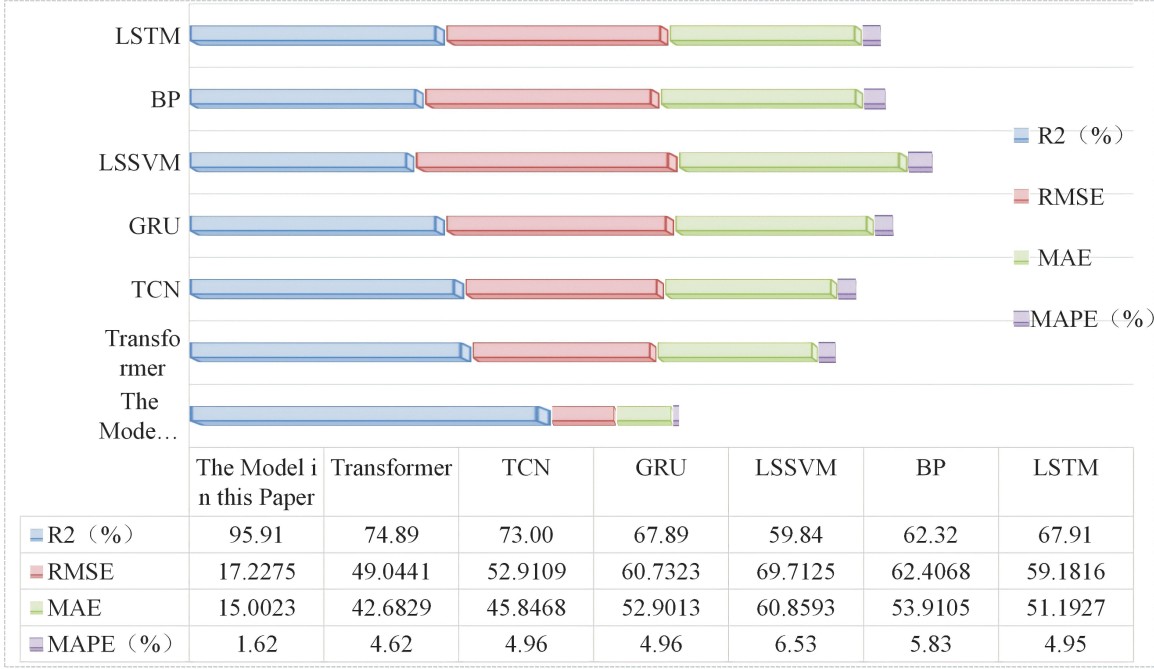

**Fig 8. Comparison of evaluation metrics results between the proposed model and other models.** Legend: Metrics: $R^2$ (determination coefficient, %), RMSE (Root- Mean Square Error), MAE (Mean Absolute Error), MAPE (Mean Absolute Percentage Error, %); Models: improved VMD-time-varying multi-model ensemble, TCN, LSSVM, BP, LSTM, Transformer, GRU.

VMD-integration model with the SMA algorithm removed; (2) The time-varying integration model with both the SMA and VMD algorithms removed; (3) A single Elman prediction model serving as an external control baseline.

However, due to journal layout constraints, this paper only uses Fig 9 to focus on presenting, in a visual format, the comparisons of prediction curves of each model in the intervals with severe fluctuations in Dataset 1—intuitively demonstrating the models' ability to capture complex fluctuation characteristics. Fig 10 conducts a systematic numerical analysis of the prediction accuracy of each model based on a multi-dimensional quantitative evaluation indicator system including RMSE, MAE, and MAPE, with the complete ablation experiment data (including prediction results and error distribution) provided in S4 File.

Through the analysis of experimental data, the proposed model demonstrates comprehensive advantages in DTR prediction. Specifically, it achieves a 95.91% $R^2$, representing increases of 6.27%, 14.01%, and 27.95% compared to the VMD-integration model, the integration model, and the Elman model, respectively. In terms of error indicators, the RMSE of the proposed model decreases by 40.7%, 53.4%, and 71.5% compared to the VMD-integration model, the integration model, and the Elman model, respectively; the MAE and MAPE indicators also show significant advantages.

First, in the data feature engineering processing stage, the non-stationarity of DTR data poses a challenge to prediction accuracy. To address this, SMA is introduced to optimize the VMD hyperparameters. Experimental data shows that the VMD-integration model (with the SMA algorithm removed) exhibits a 6.27% decrease in $R^2$, a 68.6% increase in RMSE, and a 67.9% increase in MAPE. This indicates that SMA optimization plays a decisive role in improving the quality of VMD decomposition: it can significantly reduce the non-stationarity of DTR data, provide high-quality input features for subsequent prediction models, and thus enhance prediction accuracy.

Second, the impact of VMD decomposition on the model's prediction performance is equally crucial. The integration model (with VMD further removed) performs even worse: its RMSE increases by 114.5% and MAPE rises by 111.1%

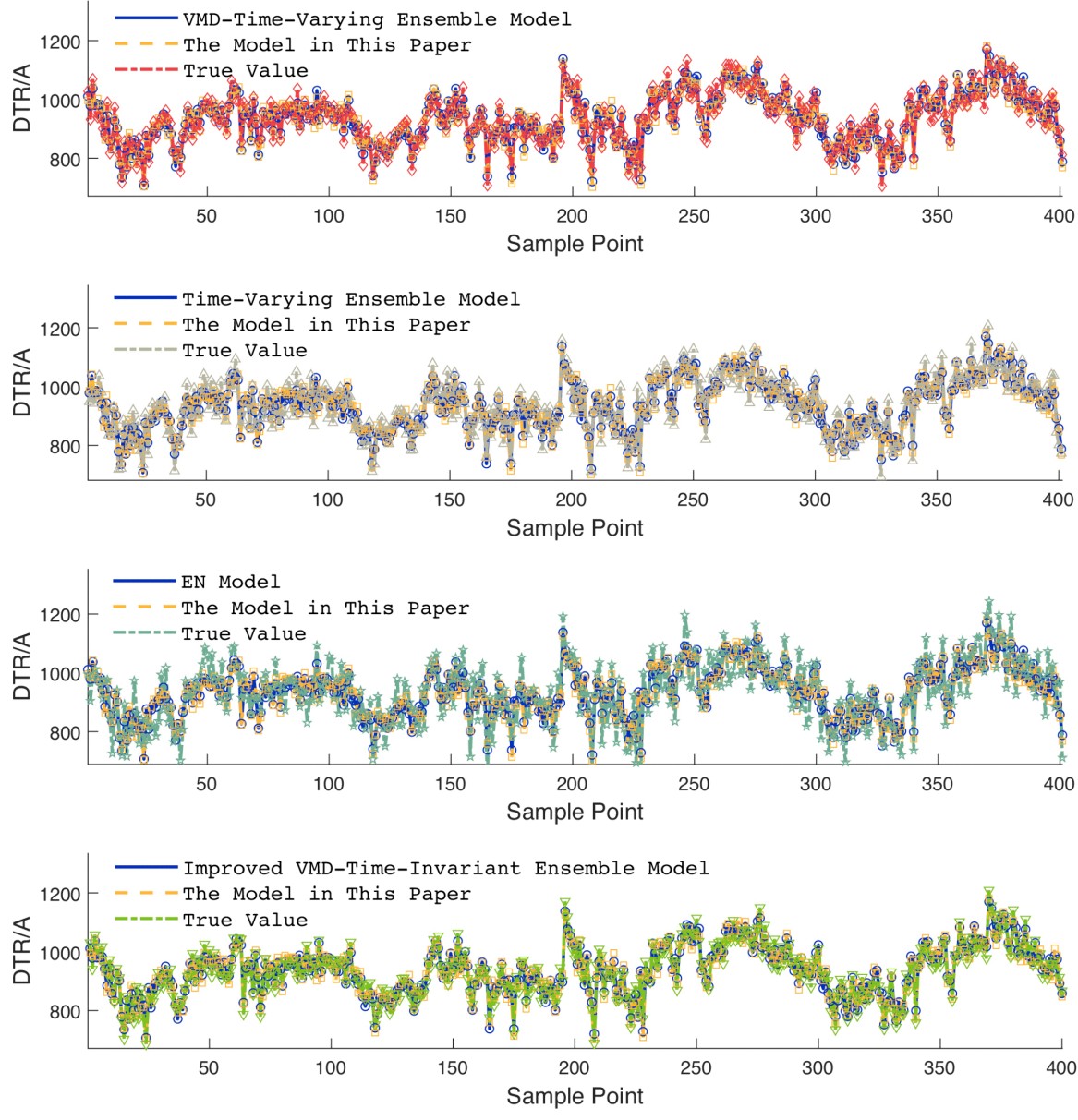

**Fig 9. DTR prediction results.**

compared to the proposed model. This data comparison strongly confirms the key value of VMD in reducing data non-stationarity.

Finally, the dynamicity of the ensemble strategy constitutes the core advantage of the proposed model in this paper. On the one hand, although the standalone Elman model excels at capturing short-term fluctuations, it suffers from insufficient stability in long-term prediction. The proposed model achieves complementary advantages through multi-model ensembling, reducing MAPE by 71.5%. compared with the standalone Elman model. On the other hand, the newly introduced time-invariant ensemble model, despite adopting an ensemble architecture, fails to adapt to the dynamic variations of DTR data due to its fixed weights. Specifically, its $R^2$ is 3.49% lower than that of the proposed model, while its root mean

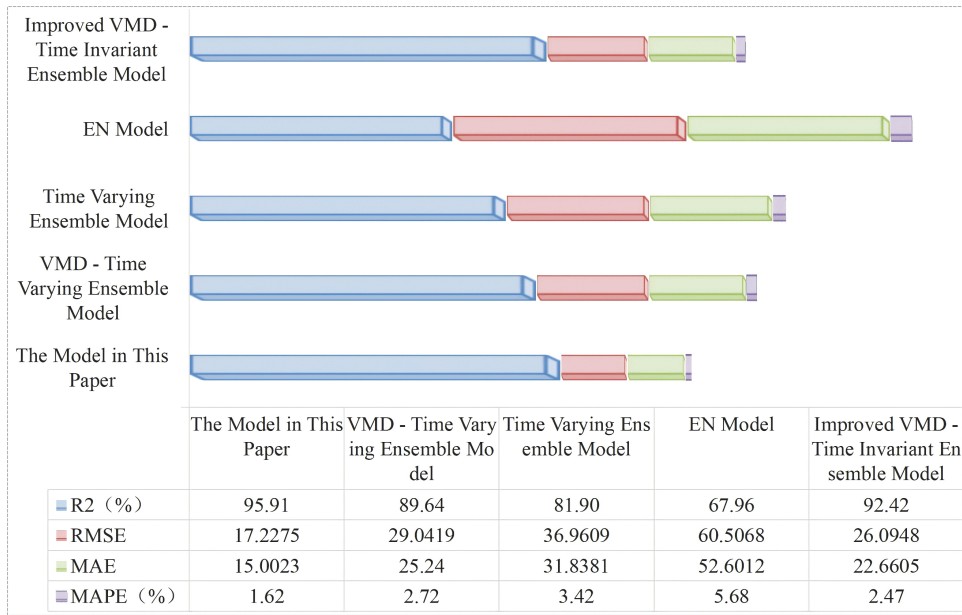

| | The Model in This Paper | VMD - Time Varying Ensemble Model | Time Varying Ensemble Model | EN Model | Improved VMD - Time Invariant Ensemble Model |
|---|---|---|---|---|---|
| R2（%） | 95.91 | 89.64 | 81.90 | 67.96 | 92.42 |
| RMSE | 17.2275 | 29.0419 | 36.9609 | 60.5068 | 26.0948 |
| MAE | 15.0023 | 25.24 | 31.8381 | 52.6012 | 22.6605 |
| MAPE（%） | 1.62 | 2.72 | 3.42 | 5.68 | 2.47 |

**Fig 10. Evaluation metrics results.**

square error (RMSE) and MAPE are 34.0% and 47.5% higher, respectively. This result verifies the value of the proposed time-varying dynamic weighting mechanism. By adjusting model weights in real time, the model can better match the temporal feature variations of the data, which serves as the key innovation point enabling the ensemble framework to achieve high-precision prediction.

In summary, the proposed model adopts a hierarchical optimization strategy integrating SMA-optimized VMD, VMD decomposition, and time-varying ensembling. This strategy fully adapts to the characteristics of DTR data from feature processing to model architecture, ultimately achieving comprehensive superiority in prediction performance. It also verifies the necessity and effectiveness of each module and the dynamic ensembling mechanism.

## Model generalization validation experiment

The generalization capability of the model across different seasons is the core metric for evaluating its engineering value. Seasonal transitions lead to significant variations in the meteorological parameters of transmission lines, which in turn induce changes in the distribution characteristics of DTR data. To verify the favorable generalization performance of the proposed model under seasonal-varying environmental conditions, comparative experiments were conducted based on DTR datasets from different seasons to systematically assess the model's generalization capability.

Both datasets were derived from a 110kV overhead transmission line (equipped with LGJ-400/20 steel-cored aluminum strands) in Zhuhai City, yet they exhibit distinct discrepancies in meteorological characteristics, data scales, and application scenarios. Detailed specifications are as follows:

Dataset 1 (Summer, July 3 to August 25, 2023) : It contains approximately 5,760 samples with a 15-minute sampling interval. The line passes through industrial zones and urban residential areas, being significantly affected by the urban heat island effect, which results in intense fluctuations of meteorological parameters: the ambient temperature ranges from 28.5°C to 38.2°C, wind speed varies between 0.3 m/s and 6.8 m/s, and the peak solar radiation intensity reaches 890 W/m². Consequently, the DTR data in this dataset features high-frequency fluctuations and short-term mutations.

Dataset 2 (Late Winter to Early Spring, January 8 to March 1, 2024): Comprising around 5,180 samples (15-minute sampling interval), the line traverses suburban and hilly areas with relatively stable meteorological conditions. The ambient temperature spans from 3.3°C to 18.7°C, wind speed ranges from 0.5 m/s to 4.2 m/s, and the peak solar radiation intensity is 720 W/m$^2$. The DTR data in this dataset is dominated by long-term periodic variations, governed by radiative heat dissipation, and its fluctuation amplitude is significantly smaller than that of Dataset 1.

The core differences between the two datasets lie in the intensity of meteorological driving factors, fluctuation frequency, and topographic environment. These differences enable effective simulation of various operational scenarios that transmission lines may encounter in practical engineering, thus providing reliable data support for the comprehensive verification of the model's generalization capability. The models selected for comparative experiments are those reported in Refs. [5,15,18], and [28], respectively. The comparative results are shown in Table 2.

From the data results, regarding the model in Reference [5], it decomposes data into components of different frequencies using the EMD algorithm, and then combines the Bayesian optimization algorithm with the BiLSTM model for prediction. In the two datasets, its RMSE values are 21.1834 and 27.8416 respectively, and MAE values are 22.7119 and 29.1864. The model shows relatively stable performance, but the error values are relatively high. This may be attributed to the frequency aliasing problem existing in EMD, which impairs the model accuracy.

The model in Reference [15] uses a BiLSTM network to capture sequence information. However, in the two datasets, its RMSE values reach 37.6972 and 40.1876, and MAE values are 32.6832 and 38.1975, indicating relatively large errors. This might be because the AL-BILSTM with a single architecture struggles to efficiently extract both short-term and long-term temporal features of DTR data simultaneously, leading to limited generalization ability.

The model in Reference [18] adopts the embedded integration of the Transformer algorithm, convolutional neural network (CNN) algorithm, and autoregressive (AR) module, aiming to address the nonlinear and linear fitting relationships of DTR data. However, in Dataset 1, the short-term mutations of DTR data and strong non-stationarity of meteorological parameters impose more stringent requirements on the model's dynamic feature capture capability. The DT-AR model in Reference [18] relies on the Transformer attention mechanism and AR module: the former tends to obscure the temporal positional sensitivity when processing time-series data, while the latter struggles to fit nonlinear abrupt relationships. Moreover, the model lacks a targeted non-stationary data processing module, resulting in a root mean square error (RMSE) of 19.9743.

The model in Reference [28] combines EMD with a BiLSTM optimized by DBO, and its performance is slightly better than the previous two models. Nevertheless, its RMSE values in the two datasets are 25.5842 and 29.7618 respectively, and MAE values are 27.6512 and 31.5764, leaving room for improvement. This could be due to the fact that the inherent defects of EMD have not been fully addressed.

**Table 2. Comparison results.**

| Dataset | Method | RMSE | MAE | MAPE | R$^2$ |
|---|---|---|---|---|---|
| **Dataset 1** | Reference [5] | 21.1834 | 22.7119 | 2.68% | 90.21% |
| | Reference [15] | 37.6972 | 32.6832 | 3.59% | 81.17% |
| | Reference [18] | 19.9743 | 20.6218 | 2.55% | 91.19% |
| | Reference [28] | 25.5842 | 27.6512 | 2.99% | 87.64% |
| | The model in this paper | 17.2275 | 15.0023 | 1.62% | 95.91% |
| **Dataset 2** | Reference [5] | 27.8416 | 29.1864 | 3.12% | 86.98% |
| | Reference [15] | 40.1876 | 38.1975 | 4.19% | 77.26% |
| | Reference [18] | 22.6742 | 22.1484 | 2.69% | 90.45% |
| | Reference [28] | 29.7618 | 31.5764 | 3.28% | 85.14% |
| | The model in this paper | 14.9913 | 13.8972 | 1.48% | 96.18% |

For the model in this paper, the RMSE values in the two datasets are 17.2275 and 14.9913, and MAE values are 15.0023 and 13.8972, all of which are significantly lower than those of other models. This advantage stems from the following: VMD enables accurate frequency separation, avoiding frequency aliasing; meanwhile, the time-varying multi-model integration framework combines the short-term recursion advantage of Elman and the long-term temporal feature extraction capability of TCN. Additionally, it optimizes prediction results through a dynamic weighting mechanism. As a result, the model demonstrates excellent generalization ability and is more adaptable to the changes in DTR data across different seasons.

To further verify the adaptability and stability of the proposed model under different scenarios, a targeted analysis of the experimental results was performed based on the characteristics of Dataset 1 and Dataset 2.

For the high-frequency and strong-fluctuation scenario (Dataset 1), In this scenario, the DTR is affected by the coupling of short-term gusts and intense solar radiation, resulting in frequent short-term mutations. Traditional models are prone to prediction deviations due to insufficient handling of non-stationarity; for instance, the model in Reference [15] yields RMSE as high as 37.6972. In contrast, the proposed model realizes adaptive decomposition of high-frequency fluctuation components via the SMA-VMD algorithm, enables the Elman model to accurately capture short-term mutation features, and dynamically adjusts the contribution weights of sub-models through the dynamic weighting mechanism. Ultimately, it achieves low-error prediction with RMSE of 17.2275 and MAPE of 1.62%, representing a 64.89% reduction in RMSE compared with the sub-optimal Transformer model. This fully demonstrates the strong adaptability of the proposed model to the high-frequency and strong-fluctuation scenario.

For the low-fluctuation and stable scenario (Dataset 2), In this scenario, the DTR is dominated by radiative heat dissipation with prominent long-term trends, which requires the model to possess a stable capability for extracting long-term features. The TCN layer of the proposed model captures long-term temporal periodic trends via dilated convolution, and the dynamic weighting mechanism adaptively tilts weights toward the TCN model. Consequently, the model achieves high-fitting prediction with a coefficient of determination $R^2$ of 96.18% and MAE of 13.8972, with the $R^2$ being 9.2% higher than that of the model in Reference [5]. This reflects the good adaptability of the proposed model to the stable trend scenario.

## Sensitivity analysis of model key parameters and computational overhead

**5.1.1. Sensitivity analysis of key parameters.** To further verify the scientificity of the model parameter configuration and the feasibility of real-time engineering applications, this section conducts a sensitivity analysis of key parameters and an evaluation of model computational overhead respectively, improving the model performance verification system from the dual dimensions of parameter robustness and engineering adaptability.

The prediction accuracy and stability of the model are highly dependent on the rational configuration of core parameters. To clarify the influence law of parameter changes on prediction results and verify the scientificity of the existing parameter selection, this section adopts the control variable method: the remaining parameters are fixed at their optimal values, and only the target parameter is adjusted within a reasonable range. The sensitivity analysis of parameters is completed by monitoring the fluctuation characteristics of the prediction index (coefficient of determination, $R^2$. The analysis focuses on three types of core parameters that directly affect prediction accuracy, namely the key parameters of the SMA, the hyperparameters of VMD, and the sliding window length of dynamic weighting.

First, the population size and maximum number of iterations of the SMA directly determine the efficiency and accuracy of VMD hyperparameter optimization. In this study, the population size was adjusted from 10 to 25 (with a step size of 5), and the maximum number of iterations was adjusted from 10 to 50 (with a step size of 10). The impact of these parameters on the final DTR prediction accuracy is shown in Table 3.

As can be seen from Table 3, when the population size is 20, the model achieves an optimal balance between accuracy and computational time. The maximum number of iterations of 10 is sufficient to reach the optimal prediction accuracy,

**Table 3. Sensitivity analysis results of SMA algorithm parameters.**

| Parameter Type | Parameter Value | $R^2$ | Conclusion |
|---|---|---|---|
| Population Size | 10 | 95.41% | Insufficient accuracy |
| Population Size | 15 | 95.44% | Insufficient accuracy with a slight increase in computation time |
| Population Size | 20 | 95.91% | Reaches optimal accuracy with controllable computation time |
| Population Size | 25 | 95.82% | Negligible change in accuracy with increased computation time |
| Maximum Number of Iterations | 10 | 95.91% | Reaches optimal accuracy with controllable computation time |
| Maximum Number of Iterations | 20 | 95.46% | Negligible change in accuracy with increased computation time |
| Maximum Number of Iterations | 30 | 95.87% | Negligible change in accuracy with increased computation time |
| Maximum Number of Iterations | 40 | 95.77% | Negligible change in accuracy with increased computation time |
| Maximum Number of Iterations | 50 | 95.42% | Negligible change in accuracy with increased computation time |

and further increasing the number of iterations will only lead to computational redundancy. It should be noted that, to ensure the fairness of the comparison between SMA and other optimization algorithms in Section 4.3 and prevent some algorithms from failing to meet the convergence criteria, the maximum number of iterations is ultimately set to 20 in this study, balancing optimization accuracy and the rationality of algorithm comparison.

Secondly, the number of decomposition components $K$ and penalty factor $\alpha$ of VMD determine the decomposition effect of DTR data. In this study, $K$ was adjusted from 3 to 12 (with a step size of 1), and $\alpha$ was adjusted from 200 to 2600 (with a step size of 500). Their impacts on the decomposition effect of DTR data and subsequent prediction accuracy are shown in Table 4 and Table 5, respectively.

As the number of decomposition components $K$ increases from 3 to 12, the component envelope entropy decreases first and then increases, reaching the minimum value of 8.39 when $K=9$; correspondingly, the coefficient of determination $R^2$ increases synchronously first and then decreases, peaking at 95.91% at $K=9$. When $K<9$, the envelope entropy decreases while the $R^2$ increases; when $K>9$, the envelope entropy rises and the $R^2$ drops. It can be concluded that $K=9$ is the optimal value, at which the data decomposition performance and prediction accuracy are both optimal.

This table presents the sensitivity analysis of the penalty factor α: as α increases from 200 to 2600, the component envelope entropy decreases first and then increases, reaching the minimum value of 8.39 when $\alpha=1200$; correspondingly, R² increases synchronously first and then decreases, peaking at 95.91% at $\alpha=1200$. When α deviates from 1200, the

**Table 4. Sensitivity analysis results of VMD hyperparameters.**

| Parameter Type | Parameter Value | Component Envelope Entropy | $R^2$ |
|---|---|---|---|
| Number of Decomposition Components $K$ | 3 | 9.21 | 82.44% |
| Number of Decomposition Components $K$ | 4 | 9.11 | 87.16% |
| Number of Decomposition Components $K$ | 5 | 8.83 | 87.43% |
| Number of Decomposition Components $K$ | 6 | 8.72 | 89.99% |
| Number of Decomposition Components $K$ | 7 | 8.51 | 92.57% |
| Number of Decomposition Components $K$ | 8 | 8.46 | 94.16% |
| Number of Decomposition Components $K$ | 9 | 8.39 | 95.91% |
| Number of Decomposition Components $K$ | 10 | 8.45 | 95.77% |
| Number of Decomposition Components $K$ | 11 | 8.43 | 95.12% |
| Number of Decomposition Components $K$ | 12 | 8.67 | 93.95% |

**Table 5. Sensitivity analysis results of VMD hyperparameters.**

| Parameter Type | Parameter Value | Component Envelope Entropy | $R^2$ |
|---|---|---|---|
| Penalty Factor $\alpha$ | 200 | 8.92 | 93.20% |
| Penalty Factor $\alpha$ | 700 | 8.65 | 94.17% |
| Penalty Factor $\alpha$ | 1200 | 8.39 | 95.91% |
| Penalty Factor $\alpha$ | 1700 | 8.72 | 94.89% |
| Penalty Factor $\alpha$ | 2200 | 8.85 | 93.76% |
| Penalty Factor $\alpha$ | 2600 | 8.79 | 92.88% |

envelope entropy increases while $R^2$ decreases. It can be concluded that $\alpha = 1200$ is the optimal value, at which both the data decomposition performance and prediction accuracy are optimal.

Finally, the sliding window length of the dynamic weighting mechanism determines the timeliness of weight calculation. In this study, the window length was adjusted from 5 to 35 (with a step size of 5), and its impacts on weight stability and prediction accuracy are shown in Table 6.

As can be seen from Table 6, when the window length is 25, the model achieves the optimal balance between prediction accuracy and computational time. When the window length is less than 20, insufficient historical data leads to excessive weight fluctuations, resulting in a prediction accuracy lower than 85%. When the window length is greater than 25, there exists a lag in weight update, along with a significant increase in computational time. Therefore, this study ultimately determines the window length as 25.

**5.1.2. Computational overhead.** The computational overhead of the dynamic weighting mechanism and the feasibility of real-time prediction constitute the core metric for evaluating the engineering value of the model. Based on the phase division of offline training and online prediction, this section conducts computational time-consuming tests and proposes optimization paths for future high-frequency data scenarios.

The computational process of the model is divided into two independent phases: offline training and online real-time prediction, and their time consumption is mutually independent. The details are as follows:

(1)  Offline training phase

This phase covers core tasks including SMA-VMD hyperparameter optimization, and training of the Elman and TCN models, which requires calling several months of historical DTR and meteorological data. In this study, a total of approximately 10,940 samples from summer and winter seasons are used; a single full-scale training takes about 1 hour. This process only needs to be completed once before model deployment, or updated slightly on a quarterly/annual basis to

**Table 6. Sensitivity analysis results of dynamic weighted sliding window length.**

| Window Length | $R^2$ | Conclusion |
|---|---|---|
| 5 | 81.64% | Insufficient historical data, leading to low accuracy |
| 10 | 82.37% | Insufficient historical data, leading to low accuracy |
| 15 | 84.32% | Insufficient historical data, leading to low accuracy |
| 20 | 92.15% | Sufficient data volume, with a significant increase in accuracy |
| 25 | 95.91% | Accuracy reaches the peak |
| 30 | 94.50% | Slight fluctuation in accuracy, accompanied by increased computation time |
| 35 | 94.59% | Slight fluctuation in accuracy, accompanied by increased computation time |

adapt to seasonal meteorological feature changes. Its time consumption is not included in the online real-time prediction process, thus not affecting the timeliness of engineering applications.

(2) Online real-time prediction phase

For DTR prediction in actual power systems, a lightweight sliding window strategy is adopted, where only recent historical data is retained within the window. Adapted to the 15-minute sampling interval in this study, the window length is set to 20 time steps, corresponding to 300 minutes of recent data. During each prediction, there is no need to recall the full-scale training data; only the model parameters completed by offline training are loaded. The grey correlation coefficient and dynamic weights are calculated based on the historical predicted values and real values within the window, and then the prediction result is output combined with real-time micro-meteorological data. Meanwhile, the window slides automatically with the prediction cycle (i.e., removing the oldest group of data and adding the latest group of data), which controls the data volume of online computation within a very small range while ensuring weight adaptability. Actual test results show that the time consumption of a single online prediction process is stably around 750 seconds, which fully meets the engineering requirements of 15-minute rolling prediction and verifies the near-real-time feasibility of the dynamic weighting mechanism.

However, with the development of micro-meteorological monitoring technology, the data sampling interval may be shortened to 5 minutes or even 1 minute in the future. The surge in data volume will place higher demands on real-time performance. To address this challenge, this study plans two types of lightweight optimization directions:

(1) Structured pruning can be performed on the model: Currently, the Elman model and TCN model contain certain redundant parameters, such as the edge convolution kernels of TCN and low-contribution neurons in the hidden layer of EN. In future work, L1 regularization can be employed to identify and remove these redundant parameters—specifically those with an impact on prediction accuracy ≤ 1%. This operation serves to reduce the number of model parameters and shorten the computational time.

(2) Parallel processing of computational tasks: The three core modules, namely VMD decomposition, dual-model prediction, and dynamic weighting, can be decomposed into independent parallel tasks. These tasks are then deployed on the multi-core hardware of edge computing nodes to enable synchronous computation of multiple modules. This approach avoids resource idleness, reduces the data transmission time between modules, and adapts to high-frequency data input scenarios.

## 6. Conclusions

(1) DTR Decomposition Based on the SMA-VMD Coupled Algorithm

To address the problem that the hyperparameters of traditional VMD rely on manual experience, leading to poor decomposition performance, the SMA is used to optimize the hyperparameters of VMD. By constructing a fitness function with the envelope entropy of DTR components as the objective, adaptive optimization of hyperparameters is achieved, and the subjectivity in parameter setting is avoided. Experimental results show that the prediction accuracy of DTR data decomposed by this coupled algorithm is improved by 14.01%.

(2) Multi-Model Integration Framework

Aiming at the problem that existing single models struggle to effectively handle the complex temporal features of DTR data, an integrated model based on the fusion of multiple single models is proposed. The integrated model captures the short-term temporal dependencies of DTR through the dynamic recursive structure of the Elman and extracts the long-term temporal relationships of DTR via the causal convolution of the TCN, thereby realizing collaborative modeling

 

of multi-scale temporal features. Experimental results indicate that the prediction accuracy of the integrated model is improved by 22.91% and 27.95% respectively compared with the single TCN model and the single Elman model.

(3) Dynamic Allocation Strategy for Time-Varying Weights

To solve the problem that traditional fixed weight allocation strategies cannot adapt to the temporal variations of DTR data, a dynamic weight allocation strategy based on time-varying grey correlation coefficients is proposed. This strategy dynamically adjusts the weights of each single model by calculating the correlation degree between the predicted values of each single model and the actual values, effectively suppressing the negative impact of outliers in individual single models on the overall prediction results. This mechanism not only improves the model's prediction accuracy but also enhances its generalization ability across different datasets.

Despite the significant technical advantages of the proposed method in this study, it still has the following inherent limitations in future large-scale engineering applications:

(1) Data dependence in complex scenarios: The high-precision prediction of the method relies on high-quality, high-frequency micrometeorological data (e.g., wind speed, ambient temperature, with a 15-minute interval). For areas with sparse meteorological monitoring stations, such as remote mountainous areas and high-altitude regions, if micrometeorological monitoring equipment at key towers cannot be supplemented, data missing or insufficient accuracy will lead to deterioration in prediction performance, thereby restricting its promotion scope.

(2) Lightweight requirement for ultra-large-scale power grids: The method consists of three core modules, namely SMA-VMD decomposition, dual-model prediction, and dynamic weighting. Although the prediction time for a single line is relatively short, in ultra-large-scale power grids with thousands of lines, the overall computational load consumption during multi-node parallel computing still needs optimization. The existing computational redundancy of edge computing nodes may fail to meet the real-time scheduling requirements under extreme conditions.

The proposed method in this study has clear application value in the dynamic capacity increase of regional power grids with complete meteorological data and complex climatic conditions. However, limited by data dependence and computational load requirements, its large-scale engineering implementation must be promoted in phases in combination with regional monitoring conditions. Future research will focus on the lightweight optimization of the model and the enhancement of robustness in data-scarce scenarios, so as to further expand the boundaries of its engineering applications.

## Supporting information

**S1 File. Raw DTR datasets (Excel format).** This file contains the complete raw data of Dataset 1 and Dataset 2.
(ZIP)

**S2 File. Iteration curve data of optimization algorithms (Excel format).** This file includes the iteration curve data of the SMA, SA, TS, PSO, WOA, GWO, APO during VMD hyperparameter optimization.
(ZIP)

**S3 File. Comparative experiment detailed data (Excel format).** This file provides the complete quantitative data of the proposed model and comparative models, including individual sample prediction values and statistical results.
(ZIP)

**S4 File. Ablation experiment detailed data (Excel format).** This file contains the full set of prediction results and indicator statistics of the ablation models and the proposed model.
(ZIP)

## Acknowledgments

The successful completion of this thesis owes much to the careful guidance and patient enlightenment of my supervisor, Professor Wubang Hao. He provided me with professional and detailed guidance on sorting out research ideas, constructing the thesis framework and revising the draft repeatedly, from which I have gained a great deal. I would like to express my sincere gratitude to Professor Hao here. I also thank all the teachers who have helped me during my studies and my university for the research support it has provided.

## Author contributions

**Conceptualization:** Siyu Yang, Wubang Hao.

**Data curation:** Siyu Yang, Wubang Hao.

**Formal analysis:** Siyu Yang.

**Funding acquisition:** Siyu Yang.

**Investigation:** Siyu Yang, Wubang Hao.

**Methodology:** Siyu Yang, Wubang Hao.

**Project administration:** Siyu Yang.

**Resources:** Siyu Yang.

**Software:** Siyu Yang.

**Supervision:** Siyu Yang.

**Validation:** Siyu Yang.

**Visualization:** Siyu Yang.

**Writing – original draft:** Siyu Yang, Wubang Hao.

**Writing – review & editing:** Siyu Yang, Wubang Hao.

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
