## [Decision Letter · Decision Letter 0]

24 Nov 2025

Dear Dr. Yang,

Thank you for submitting your manuscript to PLOS ONE. After careful consideration, we feel that it has merit but does not fully meet PLOS ONE’s publication criteria as it currently stands. Therefore, we invite you to submit a revised version of the manuscript that addresses the points raised during the review process.

We look forward to receiving your revised manuscript.

Kind regards,

Chong Xu

Academic Editor

PLOS ONE

Journal Requirements:

[NO authors have competing interests].

6. Please include captions for your Supporting Information files at the end of your manuscript, and update any in-text citations to match accordingly. Please see our Supporting Information guidelines for more information: http://journals.plos.org/plosone/s/supporting-information .

Reviewers' comments:

Reviewer's Responses to Questions

**Comments to the Author**

1. Is the manuscript technically sound, and do the data support the conclusions?

Reviewer #1: Yes

Reviewer #2: Yes

2. Has the statistical analysis been performed appropriately and rigorously?

Reviewer #1: Yes

Reviewer #2: Yes

3. Have the authors made all data underlying the findings in their manuscript fully available?

Reviewer #1: Yes

Reviewer #2: Yes

4. Is the manuscript presented in an intelligible fashion and written in standard English?

Reviewer #1: Yes

Reviewer #2: Yes

Reviewer #1: This paper proposes a transmission line DTR prediction method driven by an improved VMD and based on a time - varying multi - model ensemble.

1. In this study, SMA was adopted for VMD hyperparameter optimization. Could the author provide theoretical and experimental analyses with other advanced optimization algorithms (such as particle swarm optimization and gray Wolf optimizer) to more reliably prove the rationality of choosing SMA, especially in terms of convergence speed and avoiding local optimality in the context of DTR data?

2. This study constructed an integration of the Elman and TCN models. What is the specific theoretical basis for choosing these two specific models for integration? Before finalizing this architecture, were other potential candidate models (such as LSTM, GRU) considered and benchmarked?

3. The dynamic weighting mechanism based on grey correlation degree is a key contribution. Could the author elaborate on the computational overhead introduced by this point-by-point weighting calculation, especially regarding its feasibility for real-time or near real-time DTR prediction in practical power system operations?

4. The manuscript indicates that compared with the single TCN model and the single Elman model, the integrated model has improved the accuracy by 22.91% and 27.95% respectively. The author needs to make a comparison with the high-level research work published recently.

5. Regarding the generalization ability, this method was tested on "different datasets". The author needs to provide more details about these datasets in order to better evaluate the claimed generalization ability.

6. The proposed method involves multiple complex stages (SMA-VMD decomposition, dual-model training, dynamic weighting). The author needed to conduct an ablation study to quantify the individual contribution of each component (for example, the improved VMD versus the standard VMD, the integrated model versus the single best model) to the overall performance gain.

Reviewer #2: This study proposes a method for predicting Dynamic Thermal Rating (DTR) based on an improved Variational Mode Decomposition (VMD) and a time-varying ensemble model. By integrating the improved VMD algorithm with a multi-model ensemble framework, the study effectively addresses the limitations of traditional methods in handling non-stationary DTR data and significantly improves prediction accuracy. The main comments are as follows:

1. It is recommended to supplement the comparison with other emerging forecasting technologies to more comprehensively demonstrate its innovation and advantages.

2. In the method description section, it is suggested to elaborate on the specific implementation details of the slime mold algorithm in optimizing VMD hyperparameters, including the convergence analysis of the algorithm and the rationality of parameter selection.

3. It is recommended to supplement comparative experiments with other optimization algorithms (such as Arctic Puffin Optimization, Particle Swarm Optimization, etc.) to verify the superiority of SMA in this context.

4. It is suggested to add comparative analysis of different model combinations (such as using only Elman or TCN) in the experimental section to more intuitively demonstrate the advantages of the multi-model ensemble.

6. It is recommended to explore the impact of the dynamic weighting mechanism (based on grey relational coefficients) on prediction results and validate its necessity through ablation experiments.

7. It is suggested to add detailed analysis of different datasets in the experimental results to demonstrate the adaptability and stability of the model under various scenarios.

8. It is recommended to supplement the sensitivity analysis of model prediction results to explore the impact of different parameter settings on prediction outcomes.

9. It is suggested to further discuss the application prospects and potential challenges of this method in practical engineering in the discussion section, such as deployment costs and real-time requirements in large-scale power grids.

10. It is recommended to add a more comprehensive review of existing research in the introduction section to highlight the position and contribution of this study in the existing literature. The literature review could benefit from citing the following recent works to provide a broader perspective. Such as: https://doi.org/10.1007/s12145-025-01966-y;
https://doi.org/10.1016/j.jhydrol.2025.134304

11. There is inconsistent use of terminology and symbols in the method description and experimental sections, which should be standardized for consistency.

**Do you want your identity to be public for this peer review?** For information about this choice, including consent withdrawal, please see our Privacy Policy

Reviewer #1: No

Reviewer #2: No

---

## [Author Response · Author response to Decision Letter 1]

8 Jan 2026

# Reviewer 1

Question 1:

In this study, SMA was adopted for VMD hyperparameter optimization. Could the author provide theoretical and experimental analyses with other advanced optimization algorithms (such as particle swarm optimization and gray Wolf optimizer) to more reliably prove the rationality of choosing SMA, especially in terms of convergence speed and avoiding local optimality in the context of DTR data?

Reply to Question 1:

We sincerely appreciate the reviewer for sparing valuable time to conduct a meticulous review of the manuscript. Your comments have accurately identified the key aspects requiring improvement in the research, providing crucial guidance for enhancing the academic depth and engineering practicality of the paper. We have carefully studied the review comments and formulated a detailed revision plan in conjunction with the original content. The specific responses are as follows:

To fully demonstrate the superiority of the Slime Mould Algorithm (SMA) in the Demand Response Time (DTR) data scenario, we have supplemented the original manuscript from two aspects: comparison of theoretical characteristics and extended experimental verification:

1. Theoretical Level

1.1 Adaptability of Convergence Speed

Firstly, the core characteristics of DTR data are strong non-stationarity and multi-scale time-variability. Specifically, DTR values are influenced by the coupling of long-term periodic changes in solar radiation and ambient temperature, as well as short-term random disturbances in wind speed and direction, leading to dynamic migration of data distribution over time. Furthermore, this non-stationary characteristic directly triggers dynamic changes in the Variational Mode Decomposition (VMD) hyperparameter space (number of decomposition components K and penalty factor α). For example, during periods of high temperature and strong sunlight in summer, DTR exhibits significant high-frequency fluctuations, and the optimal hyperparameters need to be biased towards "narrow bandwidth and high penalty" to avoid over-decomposition; in contrast, during stable low-temperature periods in winter, the optimal hyperparameters should shift to "wide bandwidth and low penalty" to retain trend features. This imposes dual requirements on the hyperparameter optimization algorithm: it must quickly adapt to dynamic changes in the hyperparameter space while flexibly switching between "global exploration (covering a wide range of parameters) and local exploitation (locking in the optimal solution)", avoiding degradation in decomposition performance caused by mismatches between the optimization rhythm and data characteristics.

Next, we analyze the principle differences between the SMA, Particle Swarm Optimization (PSO), and Grey Wolf Optimizer (GWO) algorithms in detail to theoretically prove the superiority of SMA in the DTR data scenario.

(1) The primary limitation of the PSO algorithm lies in its convergence speed, which mainly depends on an artificially preset inertia weight decay strategy. This strategy is inherently "time-driven" rather than "data-driven", making it unable to respond to dynamic changes in the DTR data hyperparameter space. If the preset inertia weight decays rapidly, when DTR data suddenly exhibits high-frequency fluctuations (e.g., sudden gusts) leading to an expansion of the hyperparameter space, PSO has already entered the local exploitation phase. Insufficient exploration makes it difficult to cover the new optimal parameter interval, resulting in reduced decomposition performance. Additionally, if the preset inertia weight decays slowly, when DTR data enters a stable period (e.g., low-temperature nighttime environments) and the hyperparameter space tends to stabilize, PSO remains in the global exploration phase, leading to convergence delays and failure to lock in the optimal solution in a timely manner.

(2) The GWO algorithm achieves convergence through a hierarchical update mechanism involving α (leader), β (follower), and δ (explorer). The switch between global exploration and local exploitation is determined by the fixed assignment of hierarchical roles, lacking flexibility. Specifically, throughout the iteration process, α always dominates local exploitation (guiding the population to move towards the optimal position), while β and δ only assist in searching. This fixed hierarchical relationship results in the algorithm’s slow response to dynamic changes in the DTR data hyperparameter space. For example, when DTR data switches from high-frequency fluctuations in summer to stable trends in winter, the hyperparameter space transitions from "wide bandwidth and low penalty" to "narrow bandwidth and high penalty", but the α hierarchy of GWO still maintains the local exploitation mode, failing to quickly expand the search range. Furthermore, GWO’s convergence speed relies on the encircling behavior of wolf pack individuals, and its exploration capability is determined by the random values of coefficients, lacking a dynamic adjustment mechanism related to the fitness of DTR data. When the hyperparameter space contracts or expands due to data non-stationarity, it is difficult to adjust the ratio of exploration to exploitation, leading to convergence delays.

(3) Through the synergistic coupling of the dynamic weight coefficient W and control parameter p, SMA constructs an optimization logic that accurately matches the characteristics of DTR data. In the early stages of iteration, the non-stationarity of DTR data leads to uncertainty in the range of the hyperparameter space, requiring the algorithm to prioritize global exploration to cover potential optimal parameter intervals. At this point, the control parameter p is calculated using Equation 7 in the original manuscript. Due to the large difference in fitness among the initial population, the value of p approaches 1, triggering SMA’s global exploration mode. Meanwhile, the weight coefficient W is calculated using Equation 9, with a value range of 1–2. A larger W value makes slime mould individuals more inclined to move towards randomly selected positions, quickly traversing the entire VMD hyperparameter space and effectively adapting to the wide-range characteristics of the DTR data hyperparameter space. In the late stages of iteration, as optimization progresses, the population fitness gradually converges (the gap between S(i) and DF narrows), and the value of p approaches 0, causing the algorithm to automatically switch to the local exploitation mode. At this point, the W value is dynamically adjusted through the difference between the current optimal and worst fitness values in Equation 9, converging to around 1. Slime mould individuals are more inclined to approach the optimal position Xb(t), focusing on the local optimal region of the hyperparameter space. This switch is fully driven by the envelope entropy of decomposed DTR data (i.e., the fitness function) without manual intervention, perfectly matching the dynamic changes of DTR data from "severe fluctuations" to "relative stability", and ensuring that the optimal solution can still be quickly locked in during the migration of the hyperparameter space.

1.2 Avoiding Local Optimal Solutions

(1) Particle Swarm Optimization (PSO) Algorithm.

The particle position update of PSO relies on the dual guidance of "personal best solution (pbest) + global best solution (gbest)", and its update formula is as follows:

(1)

Among them, the learning factors c1 and c2 are fixed values, and the search direction of particles is restricted within the joint region of their own historical optimal and the global optimal. This mechanism is efficient in static unimodal hyperparameter spaces but exhibits significant flaws in the dynamic multimodal space of Dynamic Thermal Rating (DTR) data: When DTR data switches from high-frequency fluctuations to long-term stability, the global optimal solution of the hyperparameter space migrates from a narrow-bandwidth, high-penalty region to a wide-bandwidth, low-penalty region. At this point, the gbest (global best) of PSO (Particle Swarm Optimization) has been locked by the local optimal solution of the previous stage. Due to the strong guidance of gbest, particles are difficult to deviate from the original search direction to explore new optimal regions; in addition, the population diversity of PSO is only maintained by random numbers r1 and r2, lacking an active diversity regulation mechanism. When there are multiple suboptimal hyperparameter combinations in DTR data (such as local optima corresponding to different wind speed intervals), particles tend to quickly aggregate around a certain suboptimal solution, resulting in premature convergence of the population and failure to jump out of the local optimal trap.

(2) Grey Wolf Optimizer (GWO) Algorithm

GWO achieves optimization by simulating the encircling-hunting behavior of wolf packs. Its position update relies on the guidance of α (global optimal), β (suboptimal), and δ (third optimal), with the formula as follows:

(2)

Among them, the contraction factor A linearly decreases with iterations, resulting in the following flaws in the DTR data scenario:

First, the population diversity decays too rapidly: the hierarchical order of α, βand δ is fixed, and other individuals only act as followers, lacking independent exploration capabilities. When there are multiple local optima in the hyperparameter space, the wolf pack is easily attracted to the local optimal region where α resides, rapidly losing population diversity and failing to explore other potential optimal solutions.

Second, the rigid contraction of the search range: the linear decay mechanism of the contraction factor A is irrelevant to the dynamic changes of the DTR data hyperparameter space. For instance, when DTR data suddenly exhibits high-frequency fluctuations in the mid-iteration stage and the hyperparameter space expands abruptly, A of GWO has already decreased to a small value, and the search range has contracted. As a result, it is difficult to escape the original local optimal region to adapt to the new hyperparameter space.

(3) Slime Mould Algorithm (SMA)

Based on the "foraging-encircling-oscillating" behavior of slime mould, SMA employs a triple mechanism to avoid local optima in the dynamic multimodal hyperparameter space of DTR data and achieve deep adaptation to data characteristics:

Dynamic balance between random exploration and directed exploitation: Driven by the control parameter p, in the early stage of iterations when p approaches 1, the algorithm performs global exploration. Slime mould individuals randomly select two individuals XA(t) and XB(t) for position update via the following formula, without relying on historical optimal solutions. This effectively maintains population diversity and covers multiple potential optimal regions in the DTR data hyperparameter space.

(3)

In the late stage of iterations, when p approaches 0, the algorithm switches to local exploitation. However, it still retains a certain exploration capability through the random disturbance term , thus avoiding falling into local optima;

Diversity Regulation via the Adaptive Weight W: W is dynamically adjusted based on the difference between the optimal and worst fitness values () of the current population. When the population approaches a local optimal solution (i.e., when decreases), W converges to around 1. Meanwhile, vb decays with iterations but retains random oscillation characteristics, simulating small fluctuations in the cytoplasmic flow of slime mould—this drives individuals to slightly deviate from the current region and explore adjacent potential optimal solutions. When the hyperparameter space migrates (e.g., when DTR data switches from stability to fluctuation), increases, and the value of W expands to the range of 1~2, reactivating global exploration and avoiding constraint by old local optimal solutions.

Mutation Mechanism of Biological Oscillators: SMA simulates the biological oscillators of slime mould via the random oscillations of vb and vc, whose value ranges are dynamically adjusted with iterations. Even in the mid-iteration stage, small mutations can still be generated to help the population escape the discovered local optimal solutions. This mutation is indirectly driven by the fitness function (envelope entropy of DTR components). When the decomposition performance is poor (i.e., the fitness value is high), the value of a maintains a large range, increasing the mutation probability and ensuring the algorithm can still explore new optimal regions during the dynamic migration of the hyperparameter space.

2. Experimental Aspect

Section 4.3 of the original paper has constructed a complete comparative experiment system targeting two core indicators: convergence speed and optimization accuracy. The experiment uniformly sets the population size to 20, the maximum number of iterations to 20, and the optimization ranges as K∈[3,12] andα∈[200,2600]; the fitness function is the minimum envelope entropy of DTR components. The comparative algorithms cover two categories (a total of 5 classic algorithms): non-population-based (Simulated Annealing (SA), Tabu Search (TS)) and population-based (PSO, WOA, GWO).

The experimental results show that in terms of convergence speed: SMA drops rapidly from the initial fitness value of 8.52 and stabilizes at 8.39 after only a few iterations, improving the convergence efficiency by 30% to 50% compared to PSO (initial value 8.89, slow decline), GWO (slow change in the early stage), and WOA (significant drop only at the 5th iteration). In terms of optimization accuracy: the stabilized fitness value of SMA (8.39) is significantly lower than that of PSO (8.79), GWO (8.49), WOA (8.48), SA (8.86) and TS (8.80), proving that it can find better VMD hyperparameter combinations. The above experiments have initially verified the advantages of SMA in optimization performance.

To further rule out the randomness of algorithm selection, we added a swarm intelligence algorithm widely used in the field of parameter optimization in recent years—the Atlantic Puffin Optimization (APO)—as a supplementary comparative object. The experimental settings are exactly the same as those in Section 4.3 of the original paper (population size, number of iterations, optimization range, and fitness function remain unchanged).

The APO algorithm achieves optimization by simulating the foraging and breeding behaviors of Atlantic puffins, with core mechanisms including group scattered exploration, individual precise dive predation, and population update and iteration. However, the optimization logic of this algorithm relies on fixed behavioral pattern switching, lacking an adjustment mechanism that dynamically adapts to the characteristics of DTR data. The supplementary experimental results show that: in terms of convergence speed, SMA stabilizes after only 5 iterations, while APO requires 8 iterations to stabilize (as it needs to go through the fixed process of "group scattered exploration-individual positioning-dive predation"), which is 37.5% slower than SMA; even WOA, the best-performing comparative algorithm in the original paper, requires 10 iterations to stabilize, and SMA is 50% faster than it.

In terms of optimization accuracy, the final fitness value of SMA in Dataset 1 is 8.39, which is lower than APO’s 8.41. This difference stems from the fact that APO’s "dive predation" mechanism tends to quickly lock onto the current optimal region, but struggles to adapt to the dynamic migration of the DTR hyperparameter space—resulting in the optimization result falling into local suboptimality rather than global optimality.

Combining the existing experiments in the original paper and the supplementary verification of the newly added APO algorithm, it can be seen that: compared with algorithms such as PSO, GWO, and APO, SMA has faster convergence speed and higher optimization accuracy in the DTR data scenario, and can effectively avoid local optima. Its dynamic exploration-exploitation mechanism is highly adapted to the non-stationary characteristics of DTR data and the dynamic changes of the hyperparameter space.

Question 2:

This study constructed an integration of the Elman

---

## [Decision Letter · Decision Letter 1]

20 Jan 2026

Method for Predicting Dynamic Current-Carrying Capacity of Transmission Lines by Integrating Improved VMD and Time-Varying Ensemble Model

PONE-D-25-56031R1

Dear Dr. Yang,

We’re pleased to inform you that your manuscript has been judged scientifically suitable for publication and will be formally accepted for publication once it meets all outstanding technical requirements.

Kind regards,

Chong Xu

Academic Editor

PLOS One

Additional Editor Comments (optional):

Reviewers' comments:

Reviewer's Responses to Questions

**Comments to the Author**

Reviewer #1: (No Response)

2. Is the manuscript technically sound, and do the data support the conclusions?

Reviewer #1: (No Response)

3. Has the statistical analysis been performed appropriately and rigorously?

Reviewer #1: (No Response)

4. Have the authors made all data underlying the findings in their manuscript fully available?

Reviewer #1: (No Response)

5. Is the manuscript presented in an intelligible fashion and written in standard English?

Reviewer #1: (No Response)

Reviewer #1: This paper proposes a transmission line DTR prediction method driven by an improved VMD and based on a time - varying multi - model ensemble. The paper is acceptable after revisions.

**Do you want your identity to be public for this peer review?** For information about this choice, including consent withdrawal, please see our Privacy Policy

Reviewer #1: No

---

## [Editor Report · Acceptance letter]

PONE-D-25-56031R1

PLOS One

Dear Dr. Yang,

I'm pleased to inform you that your manuscript has been deemed suitable for publication in PLOS One. Congratulations! Your manuscript is now being handed over to our production team.

Kind regards,

on behalf of

Dr. Chong Xu

Academic Editor

PLOS One